# Cytotoxic Effects of Combinative ZnPcS_4_ Photosensitizer Photodynamic Therapy (PDT) and Cannabidiol (CBD) on a Cervical Cancer Cell Line

**DOI:** 10.3390/ijms24076151

**Published:** 2023-03-24

**Authors:** Radmila Razlog, Cherie Ann Kruger, Heidi Abrahamse

**Affiliations:** Laser Research Centre, Faculty of Health Sciences, University of Johannesburg, Doornfontein, P.O. Box 17011, Johannesburg 2028, South Africa

**Keywords:** photodynamic therapy (PDT), cannabidiol (CBD), cervical cancer, photosensitizer, ZnPcS_4_, HeLa, phthalocyanines

## Abstract

The most prevalent type of gynecological malignancy globally is cervical cancer (CC). Complicated by tumor resistance and metastasis, it remains the leading cause of cancer deaths in women in South Africa. Early CC is managed by hysterectomy, chemotherapy, radiation, and more recently, immunotherapy. Although these treatments provide clinical benefits, many patients experience adverse effects and secondary CC spread. To minimize this, novel and innovative treatment methods need to be investigated. Photodynamic therapy (PDT) is an advantageous treatment modality that is non-invasive, with limited side effects. The *Cannabis sativa* L. plant isolate, cannabidiol (CBD), has anti-cancer effects, which inhibit tumor growth and spread. This study investigated the cytotoxic combinative effect of PDT and CBD on CC HeLa cells. The effects were assessed by exposing in vitro HeLa CC-cultured cells to varying doses of ZnPcS_4_ photosensitizer (PS) PDT and CBD, with a fluency of 10 J/cm^2^ and 673 nm irradiation. HeLa CC cells, which received the predetermined lowest dose concentrations (ICD_50_) of 0.125 µM ZnPcS_4_ PS plus 0.5 µM CBD to yield 50% cytotoxicity post-laser irradiation, reported highly significant and advantageous forms of cell death. Flow cytometry cell death pathway quantitative analysis showed that only 13% of HeLa cells were found to be viable, 7% were in early apoptosis and 64% were in late favorable forms of apoptotic cell death, with a minor 16% of necrosis post-PDT. Findings suggest that this combined treatment approach can possibly induce primary cellular destruction, as well as limit CC metastatic spread, and so warrants further investigation.

## 1. Introduction

Cervical cancer (CC) is the fourth most frequently detected cancer in females and presents a considerable health burden to women globally [1], warranting more effective prevention, management, and control strategies [2]. Orthodox CC interventions, such as surgical removal, chemotherapy, and radiotherapy, may manifest adverse effects and are intrusive [3]. Notwithstanding notable medical advances, nearly 70% of late-stage CC sufferers experience metastasis, due to resistance to repeated therapies and disease progression [4]. This therefore necessitates the need for novel therapeutic research.

Photodynamic therapy (PDT) is an alternative treatment modality, which has presented support of CC primary abolition [5]. The mechanism of PDT is orchestrated by the interplay of red light, oxygen, and a photosensitizer (PS). The PS is activated using a red light and in turn yields cytotoxic reactive oxygen species (ROS) and singlet oxygen (^1^O_2_). The generation of these cytotoxic species induces oxidative stress, which can cause primary CC cell death via apoptosis [6]. The anti-cancer effects of PDT are derived from its direct cytotoxic effects on malignant cells, damage to the tumor vasculature and induction of inflammatory reaction, which can manifest in the development of systemic immunity, but exercise negligible effects in normal cells [7].

Cancer research into the prospective use of phthalocyanines (Pcs) as PSs in PDT is favorable. This is attributed to their superior tumor passive uptake, high ROS production, and potent absorption in the 680 nm red wavelength of light [8]. Studies have reported that zinc phthalocyanines (ZnPcs) have a strong absorption in the NIR and exhibit low absorption at wavelengths between 400 and 600 nm, potentially leading to a lower skin photosensitization when exposed to sunlight [9]. Moreover, the presence of a diamagnetic central metal, such as Zn^2+^ in the Pc nucleus, seems to improve the triplet state lifetime, as well as its yield and singlet oxygen yields compared to paramagnetic metals, however, the metalation is not required for its photodynamic activity [9].

A review by Carobeli et al. (2021) indicates that Pc PSs have potential as PDT pharmaceutical agents for anti-CC therapy. The authors firmly believe that Pc-based PS formulations could allow for the development of lead PDT compounds for the primary treatment of CC [10]. However, this review went on to note that even though Pcs PSs have excellent optical properties for in vitro PDT, the applications of unsubstituted Pc PSs within in vivo and clinical applications have a shortfall [10]. Since, unsubstituted Pc PSs are hydrophobic planar molecules, they have poor solubility and tend to aggregate in vivo, limiting their abilities to effectively reach target tissues for desired therapeutic outcomes [10]. Nonetheless, Pc PSs exhibit unique chemical structures, which allow for the introduction of various peripheral (macrocycle) and axial (central metal ion coordination) substitutions, which control the tendency for aggregation, pharmacokinetics, biodistribution, and solubility, as well fine-tuning of near-infrared spectroscopy absorbance [10,11]. Typical substitutions to improve solubility and reduce the aggregation of Pc PSs include sulfonation of the periphery of its macrocycle to make it more hydrophilic [12]. The advantage of increasing solubility of Pc PSs through sulfonation is that they can sometimes allow for direct biological administration, without the need for an additional carrier [11,13]. These attempts have led to the development of easy synthetic routes for the new generation of Pcs [10,11,14].

The potential of sulfonated Pcs for clinical PDT has prompted a search for alternate synthetic approaches to yield well-characterized compounds as single isomeric products [12]. Hydrophilic, sulfonated ZnPcS*_n_* PSs have received particular attention as PDT agents over the years, since sulfonation of the substituted ZnPc enhances its solubility to aggregate less and promote its uptake in tumor tissues, with improved in vitro and in vivo PDT outcomes [12]. However, the degree of sulfonation can also make Pc PSs bulky for direct tumor localization, and so, occasionally, the incorporation of a target nanocarrier is necessary. However, as long as the Pc PS molecules remain in their monomeric form, the quantum yield for singlet oxygen will remain unaffected [12,14,15]. 

Zinc phthalocyanine tetrasulfonate (ZnPcS_4_) has been designed as an amphiphilic PS agent to promote uptake, as well as circumvent any issues associated with solubility and aggregation [16]. Studies by Pashkovskaya et al. (2007) and Montaseri et al. (2022) noted that Pc complexes, such as anionic ZnPcS_4_ PSs, can efficiently bind to cancer tumor phospholipid membranes, through metal–phosphate coordination [16,17]. Therefore, ZnPcS_4_ PSs can spontaneously accumulate in tumors through an enhanced permeability and retention (EPR) effect, due to the binding of its tetrasulfonated groups to cancer cell membranes (which have high phospholipid contents), and so allow for passively selective accumulation [16,17,18,19].

Studies by Hodgkinson et al. (2017) demonstrated that sulphonated ZnPc, containing a mixture of differently sulfonated derivatives (called ZnPcS_mix_), was an effective PS within in-vitro-cultured HeLa cells, when treated with 4 J/cm^2^ at a wavelength of 673 nm, producing 50% cytotoxicity [20]. The study also noted that the PS was located in the cytoplasm and perinuclear region of HeLa cells [20]. More recently, a study performed by Pola et al. (2020) investigated the PDT effects of a disulfonated zinc phthalocyanine (ZnPcS_2_) PS within in vitro HeLa cells, and noted that the PS was capable of subcellular localization and that this PDT inhibited mitochondrial respiration in hypoxic conditions [21]. Additionally, Brozek-Pluska et al. (2020) analyzed the fluorescence/Raman signals of ZnPcS_4_ PS within in vivo human normal versus cancerous colon tissue samples, and demonstrated that this PS had a lower affinity for normal tissues [15]. Moreover, this same study reported that ZnPcS_4_ PS had a monomer region signal, confirming its ideal PS properties within PDT applications [15]. Lastly, this study noted that lower concentrations of ZnPcS_4_ PS were capable of endoplasmic reticulum (ER) localization, with mid concentrations capable of mitochondrial and/or Golgi apparatus lysosome localizations, while reporting preferential localization in the nucleus of colon cancer tissues at higher concentrations [15]. This was of importance, since the nucleus is the largest cellular organelle, which stores genetic information, and should PDT-induced singlet oxygen be able to destroy it in cancer tissues, enhanced therapeutic outcomes can be achieved [15]. Similarly, in vitro studies performed by Chekwube et al. (2020) successfully investigated the phototoxic effectiveness of ZnPcS_4_ PS in MCF-7 breast cancer cells, with significant cytotoxic effects reported [22]. 

According to Rak et al. (2019), other options to increase the bioavailability and solubility of metallo-phthalocyanine (MPc) PSs in cancer PDT to translate it into future in vivo and clinical trial research is by using various carrier delivery systems, such as nanoparticles (NPs), nanoemulsions, and liposomes, among others [13]. De Toldeo et al. (2020) demonstrated the improved uptake of ZnPcS_4_ PS when it was encapsulated in poly (lactic acid-glycolic acid) (PLGA) NPs and research by Naidoo et al. (2019) showed excellent active targeting subcellular uptake, with enhanced PDT treatment outcomes when conjugating ZnPcS_4_ PSs and an antibody PLGA gold NP delivery carrier, within in vitro melanoma cancer cultured cells, suggesting that both carriers could potentially serve as bioactive models [23,24]. Similarly, studies by Simelane et al. (2021) and Montaseri et al. (2022) showed improved subcellular localization and enhanced ZnPcS_4_ PS PDT treatment outcomes within in-vitro-cultured colorectal cancer cells, when conjugating to an antibody actively targeted PLGA-loaded gold NP or loading it in core/shell Ag@mSiO_2_ NP with folic acid, respectively [16,25]. In a study performed by Portilho et al. (2013), an albumin nanosphere (AN), containing ZnPcS_4_ PSs, was developed. Results reported excellent PDT anti-tumor activity within in vivo Swiss albino mice, using an Ehrlich solid tumor as an experimental model for breast cancer [26]. Intratumorally, the ZnPcS_4_-AN was capable of mediating PDT to refrain tumor aggressiveness, as well as induce regression [26]. Moreover, the use of this ZnPcS_4_-AN-mediating PDT exposed anti-neoplastic activity, such as that obtained while using intra-tumoral conventional chemo-Dox therapy [26]. More recently, Dias et al. (2022) investigated targeted liposomes (ITLs) encapsulating ZnPcS_4_ PSs in vivo interstitially, to attempt to bring this platform closer to clinical investigations [27]. The key findings from this study were that the ZnPcS_4_ PS did not elicit notable systemic toxicity in zebrafish and chicken embryos, and in human tumor breast cancer xenografts, it produced a significant tumor reduction. However, it did report skin phototoxicity in mouse models [27]. This study concluded that more effective and safer carrier delivery models must be developed to integrate this PS into a comprehensive tumor-targeting and delivery platform, so that future in vivo research can possibly translate into clinical applications [27].

The utilization of ZnPcS_4_ PS within PDT has, thus, been extensively researched in the cancers mentioned above, and has presented some promising results; however, it has not been researched within in vitro or in vivo models for possible shortcomings in CC. It was therefore decided to investigate this PS within this novel study, to provide a platform for future clinical development. Researchers do note that there is still a lot of in vitro work to be done to improve ZnPcS_4_ PS PDT in relation to its penetrability to tumor tissues, selectivity, stimulating methods, as well as promote its overall ability to overcome tumor hypoxia microenvironment to translate its use into future in vivo models. 

Moreover, even though research has reported that PDT has emerged as a possible, effective, and tolerable treatment approach for the management of primary CC, it also requires improvement in terms of investigating combinative PDT treatments to stimulate specific immune responses to eliminate secondary spread [7]. Overall, accumulating evidence indicates that the therapeutic efficacy of PDT, relies on its capacity to influence tumor–host interaction, and so by tipping the balance toward the activation of an immune response specific for malignant cells, it can eradicate metastasized cancer [28]. Therefore, to make more robust conclusions about ZnPcS_4_ PS PDT real clinical translational potential, investigations in combination with other treatments are needed. A study by Shams and colleagues (2015) suggested a need to develop PDT regimens which eradicate primary tumor growth, as well as inhibit metastasis through host immune system stimulation [29]. Chota et al. (2022) investigated the in vitro cell death mechanisms induced by *Dicoma anomala* root extract in combination with ZnPcS_4_-PS-mediated PDT in A549 lung cancer cells. This combination therapy confirmed the cytotoxic and anti-proliferative effects of *Dicoma anomala* extracts in monotherapy and in combination with ZnPcS_4_-mediated PDT, through apoptosis and the upregulation of p38, p53, Bax, caspase 3, 8, and 9 apoptotic proteins, suggesting that combinative therapies are worth exploring [30]. 

Cannabinoids are a group of naturally occurring metabolites located abundantly in the *Cannabis sativa* L. plant [31]. Research evidence denotes that the pharmacological anti-cancer therapeutic potential of the cannabinoid derivative, known as cannabidiol (CBD), from this plant can inhibit tumor cell growth and proliferation [32]. The efficacy of CBD has been attributed to its capability of targeting several cellular pathways which control tumorigenesis via the stimulation of various immune system intracellular signaling pathways [31]. The evidence for CBD’s various cancer therapeutic applications were reviewed by Zhelyazkova, Kirilov, and Momekov (2020), and its anti-proliferative and anti-invasive actions were highlighted, noting its capability to induce autophagy-mediated cancer cell death with its inherent chemotherapeutic abilities to prevent secondary spread [33].

More recently, Nkune, Kruger, and Abrahamse (2022) also reported encouraging results, demonstrating that utilizing antibody-targeted PLGA-loaded gold NP ZnPcS_4_ PS carrier as a PDT treatment for in-vitro-cultured colorectal cancer in combination with CBD allows for targeted primary tumor destruction, as well as activation of specific immune responses, which limit metastasis [34]. Lukhele and Motadi (2016) reported that CBD was able to stimulate specific cellular responses that aid its anti-cancer efficiency within in-vitro-cultured CCs, since it lessens their proliferation, and so eliminates spread [35]. Therefore, to fully understand the effectiveness of primary PDT CC treatment strategies, especially in relation to CC’s aggressive metastatic nature, there is a great demand to further explore the cytotoxic therapeutic effects of primary CC PDT in combination with secondary CBD treatments.

Thus, the aim of this study was to investigate the PDT-mediated effect of ZnPcS_4_ PS and CBD on the survival of in-vitro-cultured HeLa cells, to determine its potential application as a combinative complementary treatment form for CC. Since this is a novel study and the combinative effect of ZnPcS_4_ PS PDT and CBD has never been investigated within in vitro CC, the intention of this research was to lay the groundwork foundation in terms of investigating its potential effectiveness. These findings will possibly allow for future investigations in relation to enhancing the delivery of this combinative treatment with various drug targets and carriers, to promote its possible in vivo effectiveness. 

## 2. Results

### 2.1. HeLa ZnPcS_4_ PS ICD_50_ PDT Irradiation and LDH Cellular Cytotoxicity Dose Response Assays

To evaluate the cytotoxicity of the ZnPcS_4_ PS PDT on HeLa cells, experimental and control groups were treated with different concentrations of ZnPcS_4_ PS (0.0625, 0.125, 0.25, 0.5, and 1 µM). After laser irradiation, culture plates were re-incubated for an additional 24 h prior to being subjected to lactate dehydrogenase (LDH) membrane damage integrity analysis. Cellular lysis was induced and 100% LDH release was noted in the positive control, whereas the cells-only control indicated 3% (±SEM 0.47) cytotoxicity, and were used for statistical comparisons. The control groups of HeLa CC cells, which received increasing concentrations of ZnPcS_4_ PS (0.0625, 0.125, 0.25, 0.5, and 1 µM) alone, without any influence of irradiation, respectively, reported an insignificant incremental average increase (±16%) in cellular cytotoxicity (Figure 1). 

The experimental groups of HeLa cells which received increasing concentrations of ZnPcS_4_ PS and irradiation, showed significant linear increases in cellular cytotoxicity when compared to their respective cells, only within control groups. Experimental groups, which received 0.0625 µM ZnPcS_4_ PS and PDT, reported 33%** (±SEM 0.82) cytotoxicity, while for experimental groups which received 0.125 µM ZnPcS_4_ PS and PDT, 47%** (±SEM 1.25) cytotoxicity was somewhat significant. However, experimental groups which received 0.25 µM and 0.5 µM ZnPcS_4_ PS, as well as PDT, produced highly significant increases of 53%*** (±SEM 0.47) and 57%*** (±SEM 0.48) in their respective cytotoxicity assays. The most significant increase in cytotoxicity of 63%*** (±SEM 0.82) was reported in experimental groups, which received 1 µM ZnPcS_4_ PS and PDT. 

### 2.2. HeLa CBD ICD_50_ Irradiation and LDH Cellular Cytotoxicity Quantitative Dose Response Assays

To evaluate the cellular cytotoxicity of CBD on HeLa cells, experimental and control groups were treated with different concentrations of CBD (0.3, 0.5, 0.7, 0.9, and 1.1 µM). The control groups of HeLa cells which received 99.8% (*v*/*v*) ethanol only, generated an incremental increase of 14% (±SEM 0.34) in cellular cytotoxicity. The experimental group, which received 0.3 µM CBD and irradiation, reported 43%** (±SEM 1.05) cytotoxicity, which was somewhat significant. However, experimental groups which received 0.5 µM and 0.7 µM CBD, as well as irradiation, denoted significantly increased values of 52%** (±SEM 0.75) and 56%** (±SEM 0.78), with reference to their respective cytotoxicity assays. The most significant increases in cytotoxicity of 57%** (±SEM 0.86) and 62%*** (±SEM 0.42) were reported in the experimental groups which received 0.9 µM and 1.1 µM CBD plus irradiation, respectively. Similar overall average cytotoxicity results were reported for control groups consisting of cells plus CBD of ±53%** (Figure 2). 

### 2.3. Qualitative Subcellular Localization Immunofluorescent Staining Confirmation of ZnPcS_4_ PS Uptake in HeLa and WS1

The HeLa-cells-only control group, which received no forms of treatment, was utilized as the control comparator for result interpretation outcomes (Figure 3). 

From these images, cellular green membrane staining, and blue cellular nuclei of the HeLa-cells-only were identified and collated into an overlay image. Control groups of HeLa cells which received 0.125 µM ZnPcS_4_ PS, and experimental groups which received 0.125 µM ZnPcS_4_ PS + 0.5 µM CBD, also had their green cellular membrane proteins and blue-stained nuclei, as well as their red fluorescent ZnPcS_4_ PS signal identified clearly. When these three images were overlayed, it was possible to potentially identify and locate where the ZnPcS_4_ PS had localized within HeLa cells (Figure 3). 

The WS1-cells-only control group, which received no forms of treatment, was utilized as the benchmark comparator for result interpretation outcomes. From these images, the cellular green membrane staining, and blue cellular nuclei of the WS1 fibroblast cells only were identified and collated into an overlay image. Control groups of WS1 fibroblast cells which received 0.125 µM ZnPcS_4_ PS, and experimental groups which received 0.125 µM ZnPcS_4_ PS + 0.5 µM CBD, also had their green cellular membrane proteins and blue-stained nuclei, as well as their red fluorescent ZnPcS_4_ PS signal identified clearly. When these three images were overlayed, it was possible to potentially identify and locate where the ZnPcS_4_ PS had localized within the WS1 fibroblast cells (Figure 4).

### 2.4. HeLa Flow Cytometry Cell Death Pathway Quantitative Analysis of ZnPcS_4_ PS and CBD PDT/Irradiation in Combinative Assays

With reference to Figure 5, the percentage of different stages of cell death, using the flow cytometry Annexin V-FITC/PI staining method on various HeLa control and experimental groups within ZnPcS_4_ PS and CBD PDT/irradiation combinative assays, has been shown.

The cells-only positive control, which received no forms of treatment produced a 96% (±SEM 0.68) viable population of cells, with negligible cell death. The positive Annexin-V-FITC-stained control of apoptotic-induced cell death denoted only 12% (±SEM 0.45) of viable cells, a significant 49% (±SEM 0.48) of early apoptosis and 37% (±SEM 0.41) of late apoptosis, as well as a minor 2% (±SEM 0.51) of necrotic cell death. The negative PI-stained control of necrotic-induced cell death denoted only 10% (±SEM 0.57) of viable cells, a minor 8% (±SEM 0.61) of early apoptosis, and 6% (±SEM 0.56) of late apoptosis; however, a significant 76% (±SEM 0.49) necrotic cell death was also observed. Thus, these positive controls were considered as acceptable standards for result comparisons and statistical data interpretations. 

The control group of HeLa cells plus irradiation reported a substantial 92% (±SEM 0.92) of viable cells, without any other significant forms of cell death. The control group of HeLa cells, which received 99.8% (*v*/*v*) of ethanol only, reported a majority of 87% (±SEM 0.53) of viable cells, without any other significant forms of cell death. 

The control group of HeLa cells which received 0.125 µM ZnPcS_4_ PS alone, reported a noteworthy 84% (±SEM 0.96) of viable cells, without any other significant forms of cell death. The HeLa cells which received 0.125 µM ZnPcS_4_ PS and irradiation reported a significant reduction in viability, whereby only 52%* (±SEM 0.87) of cells were noted as viable. Furthermore, 15% (±SEM 0.92) of these cells were undergoing early apoptosis, and an even more significant 24%** (±SEM 0.85) were in a late apoptotic form of cell death, with an incremental amount of 9% (±SEM 0.56) necrosis. 

The control and experimental groups of HeLa cells which received 0.5 µM CBD alone, with or without laser irradiation, respectively, reported similar significant forms of cell death. The control group which received 0.5 µM CBD without irradiation noted a significant decrease of only 46%** (±SEM 0.44) of cells being viable, as well as a significant increase of 38%** (±SEM 0.46) of cells undergoing late apoptosis, with incremental amounts of early apoptosis and necrosis. Similarly, the experimental group which received 0.5 µM CBD with irradiation noted a significant decrease of only 50%** (±SEM 0.56) of cells being viable, as well as a significant increase of 42%** (±SEM 0.62) of cells undergoing late apoptosis, with incremental amounts of early apoptosis and necrosis. 

The experimental groups of HeLa cells which received 0.125 µM ZnPcS_4_ PS plus 0.5 µM CBD, without PDT, reported a significant reduction, whereby only 35%** (±SEM 0.76) of cells were viable. Furthermore, 11% (±SEM 0.66) of these cells were undergoing early apoptosis and an even more significant 45%** (±SEM 0.72) of these cells were in a late apoptotic form of cell death, with an incremental amount of 9% (±SEM 0.55) necrosis. 

In contrast, HeLa cells which received 0.125 µM ZnPcS_4_ PS plus 0.5 µM CBD and irradiation reported the most highly significant and favorable forms of cell death, with only 13%*** (±SEM 1.04) of the cells being viable. A further 7% (±SEM 0.98) of the cells were in early apoptosis and a confounding 64%*** (±SEM 0.93) were in late forms of favorable apoptotic cell death, with a minor 16% (±SEM 1.12) of necrosis post-PDT. In comparison to the experimental groups of HeLa cells which received 0.125 µM ZnPcS_4_ PS plus 0.5 µM CBD without PDT, the percentages of cellular viability were significantly decreased (35%** vs. 13%**) and the percentages of late apoptotic cell death were also significantly increased (45%** vs. 64%***). 

### 2.5. WS1 Normal Fibroblast Flow Cytometry Cell Death Pathway Quantitative Analysis of ZnPcS_4_ PS and CBD PDT/Irradiation in Combinative Assays

With reference to Figure 6, the percentage of different stages of cell death using the flow cytometry Annexin V-FITC/PI staining method on various WS1 normal human fibroblast control and experimental groups within ZnPcS_4_ PS and CBD PDT/irradiation combinative assays, are shown. 

The cells-only positive control noted a 98% (±SEM 0.51) viable population of cells, with negligible cell death. The positive Annexin-V-FITC-stained control of apoptotic-induced cell death noted only 10% (±SEM 0.73) of viable cells, a significant 34% (±SEM 0.81) of early apoptosis and 52% (±SEM 0.79) of late apoptosis, as well as a minor 4% (±SEM 0.84) necrotic cell death. The negative PI-stained control of necrotic-induced cell death noted only 8% (±SEM 0.32) of viable cells, a minor 6% (±SEM 0.46) of early apoptosis and 8% (±SEM 0.39) of late apoptosis; however, a significant 78% (±SEM 0.41) necrotic cell death was also observed. 

The control group of WS1 cells plus irradiation reported a substantial 96% (±SEM 0.60) of viable cells, without any other significant forms of cell death. The control group of WS1 cells which received 99.8% (*v*/*v*) of ethanol only, reported a majority of 88% (±SEM 1.01) of viable cells, without any other significant forms of cell death. 

The control group of WS1 cells which received 0.125 µM ZnPcS_4_ PS without any influence of irradiation, reported a noteworthy 85% of viable cells, without any other significant forms of cell death. The control groups of WS1 (±SEM 0.44) cells which received 0.125 µM ZnPcS_4_ PS and irradiation, did not report significant forms of cell death, however, in comparison to normal cells, only 74% (±SEM 0.63) remained viable, with an 8% (±SEM 0.67) increase in late apoptosis. 

The control and experimental groups of WS1 cells which received 0.5 µM CBD with or without laser irradiation, respectively, reported no significant forms of cell death. The control group which received 0.5 µM CBD without irradiation noted 87% (±SEM 0.58) of cells being viable, whereas the experimental group which received 0.5 µM CBD with irradiation similarly noted 86% (±SEM 0.87) of cells being viable. 

The experimental groups of WS1 cells which received 0.125 µM ZnPcS_4_ PS plus 0.5 µM CBD without irradiation also reported no significant reduction in cell viability, with incremental amounts of cell death. Similarly, the same results were reported in control groups which received 0.125 µM ZnPcS_4_ PS alone or 0.5 µM CBD alone. 

The experimental group of WS1 cells which received 0.125 µM ZnPcS_4_ PS plus 0.5 µM CBD with laser irradiation noted no significant forms of cell death and 73% (±SEM 0.93) of the cell population remained viable. However, in comparison to the cells-only control group, there was a higher population of early (10% ±SEM 1.03) and late apoptosis (13% ±SEM 0.99) being noted. Similar findings were reported in control groups which received 0.125 µM ZnPcS_4_ PS and PDT, whereby 12% (±SEM 0.77) of early and 8% (±SEM 0.47) of late apoptosis was noted. Subsequently, control groups of WS1 cells which received 0.5 µM CBD with irradiation, respectively, showed slightly less (7% ±SEM 0.86) early apoptosis and 4% (±SEM 0.79) of late apoptosis. 

Nonetheless, even though the cell death was insignificant, since minor early apoptosis/autophagy was observed, there was a possibility that WS1 post-PDT could recover, which required corroboration in ATP proliferation assays.

### 2.6. HeLa Adenosine Triphosphate (ATP) Quantitative Cell Proliferation Analysis of ZnPcS_4_ PS and CBD PDT/Irradiation in Combinative Assays

With reference to Figure 7, the percentage of cell proliferation using the ATP illumination method among various HeLa control and experimental groups, within ZnPcS_4_ PS and CBD PDT/irradiation combinative assays, has been shown. 

The cells-only positive control, which received no forms of treatment, noted a 100% (±SEM 1.07) viable population of cells capable of cellular proliferation, and was thus considered an acceptable standard for result comparisons and statistical data interpretations.

The control groups of HeLa cells which received irradiation, 99.8% (*v*/*v*) of ethanol or 0.125 µM ZnPcS_4_ PS only, noted no significant decreases in cellular proliferation. However, the control groups of HeLa cells which received 0.125 µM ZnPcS_4_ PS and laser irradiation reported a significant decrease, as only 20%*** (±SEM 0.82) of the cell population post-PDT treatment was able to proliferate. 

The control and experimental groups of HeLa cells which received 0.5 µM CBD, with or without laser irradiation, reported similar significant decreases, since only of 22%*** (±SEM 0.47) and 23%*** (±SEM 0.82) of the cell populations pre- and post-PDT treatment, respectively, were capable of proliferation. 

The experimental groups of HeLa cells which received 0.125 µM ZnPcS_4_ PS plus 0.5 µM CBD without PDT reported a significant reduction, whereby only 18%*** (±SEM 0.82) of the cells maintained their ability to proliferate, whereas HeLa cells which received 0.125 µM ZnPcS_4_ PS plus 0.5 µM CBD and PDT reported the most significant decreases in cellular proliferation, noting that only 6%*** (±SEM 0.94) of the cellular population was capable of proliferation. 

### 2.7. WS1 Normal Fibroblast ATP Quantitative Cell Proliferation Analysis of ZnPcS_4_ PS and CBD PDT/Irradiation in Combinative Assays

With reference to Figure 8, the percentage of cell proliferation using the ATP illumination method whiathin various WS1 normal human control and experimental groups within ZnPcS_4_ PS and CBD PDT/irradiation combinative assays, has been shown. 

The positive control of cells only noted a 100% (±SEM 0.98) viable population of cells, capable of cellular proliferation and so was considered as an acceptable standard for result comparisons and statistical data interpretations. 

## 3. Discussion

### 3.1. HeLa ZnPcS_4_ PS ICD_50_ PDT and LDH 

The control groups of HeLa cells which received laser irradiation at 673 nm and a fluency of 10 J/cm^2^, showed increases in cellular cytotoxicity (9%) (Figure 1). This is consistent with literature, since findings by Nkune, Kruger, and Abrahamse (2022), Mokoena, George, and Abrahamse (2019), and Chizenga et al. (2019), noted that laser irradiation alone at a 673 nm wavelength and fluency of 10 J/cm^2^ caused negligible cellular viability and cytotoxicity effects on their in-vitro-cultured colorectal, breast or CC cancer controls, respectively [5,34,36]. These findings are suggestive that at this wavelength and fluence, laser irradiation alone played no significant part in the PDT treatment outcomes of the HeLa cells. 

The control groups of HeLa cells which received increasing concentrations of ZnPcS_4_ PS alone without any influence of irradiation, respectively, reported insignificant incremental average increases (±16%) in cellular cytotoxicity (Figure 1). The outcome suggested that these varying concentrations of ZnPcS_4_ PS, when administered alone to HeLa cells (without laser irradiation excitation), remained inactive and lacked any form of dark phototoxicity. Second-generation ZnPc Pcs, such as ZnPcS_4_ PSs, have shown high stability in the absence of light [9,15,37,38,39]. 

Findings for this assay suggest that laser irradiation, with the administered ZnPcS_4_ PS in these dose concentration ranges, could become excited and sufficiently activated to produce considerable amounts of ROS and ^1^O_2_ cytotoxic species. This also implies that it is capable of significant in-vitro-cultured HeLa primary cellular destruction. Related studies have stated that since ZnPcS_4_ is a metalated second-generation Pc PS, its cytotoxic PDT treatment outcomes for cancer are often highly favorable and it allows for activation within the far-red spectral range of 634 to 674 nm. This permits for maximum tissue depth penetration and excitation, and high yields of localized PDT-induced cytotoxic species, which in turn can cause considerable primary tumor destruction [9,37,38,39].

The determined ICD_50_ ZnPcS_4_ PS concentration able to induce approximately 50% cytotoxicity within in-vitro-cultured HeLa experimental groups, was found to be 0.125 µM. This concentration was chosen since it reported 47%** cytotoxicity, as it ensured that HeLa cells remained somewhat viable post-PDT, so that their overall combinative biological and cell death effects could be measured. 

Since hydrophilic ZnPcS_4_ PS PDT primary tumor destruction has not yet been investigated within in-vitro-cultured HeLa cells, these cytotoxicity results seem promising, as the ICD_50_ results at a concentration of 0.125 µM were much lower than in previous studies by Hodgkinson et al. (2017) at 1 μM of ZnPcS_mix_ PS, and by Pola et al. (2021) at 0.5 μM ZnPcS_2_ PS [20,21]. 

### 3.2. HeLa CBD ICD_50_ Irradiation and LDH 

The control groups of HeLa cells which received laser irradiation showed slight increases in cellular cytotoxicity (10%) (Figure 2). This is consistent with findings from Chizenga, Chandran, and Abrahamse (2019), deducing that irradiation alone at a 673 nm wavelength and fluency of 10 J/cm^2^ caused negligible cellular viability or cytotoxicity effects on their in-vitro-cultured HeLa controls [5].

The control groups of HeLa cells which received 99.8% (*v*/*v*) ethanol showed 14% cellular cytotoxicity, suggesting that the CBD solvent played no substantial role in the treatment outcomes. Supportively, studies by Ramer et al. (2010) which investigated the treatment effects of CBD on cultured HeLa cells reported that CBD solvents, such as ethanol, which were individually controlled for within assays, had no contributing cytotoxic effects on the treatment outcomes either [40]. 

The control groups of HeLa cells which received linear increments of increasing concentrations of CBD alone, without any influence of PDT, respectively, reported significant average increases (±53%**) in cellular cytotoxicity. These results showed that as the concentrations of CBD increased, cellular cytotoxicity proportionally increased. The determined ICD_50_ CBD concentration, able to induce approximately 50% cell damage within HeLa experimental groups, was found to be 0.5 µM, since it reported 52%** cytotoxicity.

These significant findings mirrored the results observed within the experimental groups of HeLa cells which received the same linear increments of increasing concentrations of CBD with laser irradiation, when compared to their respective cells-only control groups. The results suggest that irradiation had no influence on the CBD treatment effects, and that the induced highly significant increases in cytotoxicity could be attributed to the therapeutic effects of CBD alone. Nkune et al. (2022) investigated the synergistic treatment effects of CBD and ZnPcS_4_ PS PDT on in-vitro-cultured colorectal cancer cells, whereby CBD individual dose response assays (0.5, 0.1, 1.5, and 2 μM) were performed on control groups which received CBD alone and experimental groups which received CBD and laser irradiation [34]. The outcomes of this study likewise noted that CBD demonstrated a dose-dependent inhibitory effect on the in-vitro-cultured colorectal cancer cells, regardless of whether laser irradiation (within the same parameters as utilized in this study) was present or absent [34]. Moreover, Lukhele and Motadi (2016), who investigated the effects of CBD alone on in-vitro-cultured CC HeLa cells, reported a staggering 43.3% induced cytotoxic cell death, when 3.2 µg/mL of CBD was administered with a non-recovery of the dead cell population [35]. This study went on to attribute these substantial cytotoxic outcomes to the various active secondary metabolites that CBD contains, which may over-stimulate various cellular signaling pathways to generate anti-tumor responses [35].

### 3.3. Qualitative Subcellular Localization Immunofluorescent Staining Confirmation of ZnPcS_4_ PS Uptake

As can be observed from both the control groups of HeLa cells in Figure 3 which received 0.125 µM ZnPcS_4_ PS, and experimental groups which received 0.125 µM ZnPcS_4_ PS + 0.5 µM CBD, the amount of ZnPcS_4_ PS uptake appeared fairly equal, and so it can be assumed that CBD had no effect upon the absorption of ZnPcS_4_ PS into the CC HeLa cells. This correlates with the study by Nkune et al. (2022), which investigated the combinative treatment effects of ZnPcS_4_ PS and CBD on colorectal cancer cells [34]. They noted that the absorption of the ZnPcS_4_ PS, with or without the addition of CBD, remained the same, and concluded that its presence would not affect the subcellular localization of this PS [34]. 

Furthermore, from both images taking the placement of the green-stained CC HeLa cell membrane and their blue-stained nucleus in overlay images into account, it can be observed that the ZnPcS_4_ PS was able to enter the cells successfully and was possibly able to mainly sub-localize in the cytoplasm, rather than in the cellular membrane. Carobeli et al. (2021) also stated that metallo-Pc PS formulations, which exhibit higher capacities to permeate the cytoplasmic membrane and internalize within the cell cytoplasm, reported far more effective PDT outcomes [10]. This is attributed to ROS having a short half-life, and so when a PS becomes excited, the closer it is internalized within cancer cell contents and the nuclei, the far better the intra-tumoral destruction is [10]. Pazos and Nader (2007) investigated the effect of PDT and various PSs on the extracellular matrix and associated components of cancer cells, and stated that only hydrophilic PSs can rapidly and passively diffuse into plasmatic membranes, while more polar PSs tend to only be internalized via endocytosis or by assisted transport serum lipids and proteins [41]. These qualitative outcomes are suggestive that since this ZnPcS_4_ PS was able to permeate the cytoplasmic membrane of HeLa cells, it would have the opportunity to sub-localize within the mitochondria and lysosomes of HeLa cells, and so possibly be able to induce the most desired mechanism of programmed cell death in PDT, which is apoptosis and not necrosis [42]. 

Necrosis is described as a rapid and relatively broad mechanism of cell death, characterized by vacuolization of the cytoplasm and cell membrane destruction. A localized inflammatory reaction is subsequently induced due to the release of cytoplasmic content and pro-inflammatory mediators in the extracellular space [43]. High doses of PDT and/or uncontrolled photodamage also lead to the unrestrained release of biomolecules during unregulated cell death into the extracellular medium, initiating inflammatory responses in the surrounding healthy tissue, and so necrosis is generally seen as an undesirable mechanism in PDT [44]. In addition, Ali and van Lier (1999) have stated that the sulfonation of the substituted ZnPc enhances its solubility to aggregate less, and promotes its uptake in tumor tissues [12]. Moreover, research by Pashkovskaya et al. (2007), Montaseri et al. (2022), and Nkune and Abrahamse (2023) noted that anionic ZnPcS_4_ PSs can efficiently bind to tumor phospholipid membranes, through metal–phosphate coordination and spontaneously accumulate in tumors through an EPR effect. This is attributed to its tetrasulfonated groups passively high affinity toward cancer cell membranes [16,17,19]. In vivo tissue studies by Brozek-Pluska et al. (2020) noted that ZnPcS_4_ PSs have a higher affinity for colon cancerous human tissues than normal tissues, as well that lower concentrations of ZnPcS_4_ PS were capable of ER localization [15].

As can be observed from both control groups of WS1 fibroblast cells in Figure 4 which received 0.125 µM ZnPcS_4_ PS and experimental groups which received 0.125 µM ZnPcS_4_ PS + 0.5 µM CBD, there was a slight uptake of the ZnPcS_4_ PS in the normal control cells, regardless of whether CBD was present. The subcellular localized uptake of PS is an essential factor in PDT, since it determines PDT potency and the PS’s direct passive or active intracellular localization at the tumor site, which in turn can enhance overall cell death treatment outcomes [44]. However, in relation to the intensity of ZnPcS_4_ PS red autofluorescence, when compared to the same control and experimental groups of HeLa cells in Figure 3, the uptake was far lower. Avşar and colleagues (2016) investigated the intracellular uptake and fluorescence imaging potential of ZnPcS_2_ PSs in HeLa cells versus normal human fibroblast lung WI-38 cells [45]. They noted that the PS showed four times higher intracellular uptake in HeLa cells when compared to normal fibroblast cell lines [45]. The results concluded that the ZnPcS_2_ PSs showed potential for CC PDT treatment due to their ability to localize within the cytoplasmic region of CC cells more passively compared to normal human cells [45]. However, since the ZnPcS_2_ PSs did show slight absorption in the normal human control cells, when laser irradiation was applied to a localized tumor, some of the normal surrounding healthy tissues became compromised. Nevertheless, the benefits of such treatment outcomes to eradicate primary tumor cells would outweigh the negligible damage caused in normal tissues, since the laser activation of PS for effective treatment outcomes remains localized [45]. Overall, the results from this qualitative assay suggest that ZnPcS_4_ PS remained unaffected in relation to subcellular localization in the presence of CBD. Additionally, since it was able to accumulate within in-vitro-cultured CC HeLa in a more concentrated manner, than when compared to normal WS1 fibroblast cells, these findings hint that it should be capable of more localized PDT treatment outcomes. Since the above images were captured utilizing fluorescent microscopy with fixed cells, there is a possibility that they could contain fixation-induced artifacts in the distribution of ZnPcS_4_ PS, which can affect their overall resolution, and so they can only be utilized for quantitative observations with hypothetical conclusions. Further confirmatory confocal microscopy studies with live cells need to be performed and measured to confirm the possible sub-cellular localization abilities of ZnPcS_4_ PS.

### 3.4. HeLa Flow Cytometry Cell Death Pathway Quantitative Analysis 

The control group of HeLa cells plus laser irradiation reported a substantial 92% of viable cells, without any other significant forms of cell death (Figure 5). Chizenga et al. (2019) also noted that HeLa cells, when irradiated alone at the same laser parameters as used in this study, remained unaffected post-irradiation in the absence of the AlPcS_mix_ PS [5].

The control group of HeLa cells which received 99.8% (*v*/*v*) of ethanol only, reported 87% viable cells without any other significant forms of cell death, suggesting that this solvent played no substantial role within the combinative treatment outcomes. Ramer et al. (2010) also noted that CBD ethanol solvent alone within in-vitro-cultured HeLa cells had no contributing cytotoxic effects on the treatment outcomes [40].

The control group of HeLa cells which received 0.125 µM ZnPcS_4_ PS alone, without any influence of irradiation, reported a noteworthy 84% of viable cells. Related studies substantiate these findings, whereby ZnPcS_4_ PSs have demonstrated high photochemical stabilities in the absence of light [15,38,39].

The control groups of HeLa cells which received 0.125 µM ZnPcS_4_ PS and irradiation reported a significant reduction in viability (52%* of cells were noted as viable). These results confirmed that the chosen ICD_50_ 0.125 µM ZnPcS_4_ PS concentration was capable of red laser light excitation, as well as yielding sufficient ROS and ^1^O_2_ cytotoxic species to induce an overall 48% cell death (of which a significant 24%** were, late apoptotic). 

PDT can induce three types of cell death in cancer: type I (autophagy/early apoptosis), type II (late apoptosis), and type III (necrosis) [46]. When PDT is applied at high concentrations of PS or strong irradiation doses, cancer cells tend to undergo rapid and unscheduled (or accidental) necrotic death, which is known to be non-immunogenic and unfavorable, as it provokes the immune system to cause unwanted inflammation [46]. In contrast, PDT applied at very low concentrations of PS or low irradiation doses to cancer cells tends to undergo autophagy or early apoptosis, which can simultaneously induce pro-survival and unwanted recovery [46]. Thus, the most favorable form of PDT cell death, when optimal concentrations of PS and irradiation doses are applied, is late apoptosis, since it is a programmed cell death pathway which triggers executioner caspases, resulting in chromatin and nuclear shrinkage, DNA fragmentation, as well as cellular fragmentation, without any unwanted inflammatory responses from which cancer cells cannot recover [46].

Since control groups of HeLa cells which received 0.125 µM ZnPcS_4_ PS and laser irradiation reported 15% of cells undergoing early apoptosis and a significant 24%** of cells in a late apoptotic form of cell death, with an incremental amount of 9% necrosis, it can be safely assumed that this treatment form was capable of primary CC cell death, with minimal inflammation. Moreover, these outcomes support the subcellular localization findings, whereby it was reported that 0.125 µM ZnPcS_4_ PS were significantly localized in HeLa cells, signifying that this PS was capable of subcellular localization and favorable apoptotic forms of cell death post-PDT. Since control groups of HeLa cells which received 0.125 µM ZnPcS_4_ PS and irradiation noted significant amounts apoptotic cell death post-PDT, these results are suggestive of the ZnPcS_4_ PS ability to permeate the cytoplasmic membrane of HeLa cells and sub-localize within the mitochondria and lysosomes. Metallo-Pc PS formulations which exhibit higher capacities to permeate the cytoplasmic membrane and internalize within the cell cytoplasm report far more effective PDT outcomes, and since ROS has a short half-life, when a PS becomes excited, the closer it is internalized within a cancer cells contents and nuclei, the far better the controlled intra-tumoral apoptotic destruction is [10,42]. Studies by Pola et al. (2021), and Pazos and Nader (2007), which investigated the effect of PDT and various PSs on the extracellular matrix and associated components of cancer cells, stated that only hydrophilic PSs can rapidly and passively diffuse into plasmatic membranes, resulting in apoptotic cell death, while more polar PSs tend to only be internalized via endocytosis or by assisted transport serum lipids and proteins, and cause unfavorable inflammatory necrotic cell death [21,41].

These results confirmed that the chosen ICD_50_ 0.5 µM CBD concentration was capable of cytotoxic cell death, since it induced an overall approximate 50% cell death (of which a significant 38%** pre-laser and 42% post-laser irradiation treatments were late apoptotic). The Lukhele and Motadi (2016) study attributed the significant increases of cell death to the various secondary metabolites in CBD, which stimulate various cellular signaling pathways to generate late apoptotic and anti-tumor proliferation responses in cancer cells [35]. Comparatively, strong apoptotic forms of cell death noted within in-vitro-cultured Hela cells, when treated with CBD, are credited to the fact that CC has strong expression patterns of CB1 and CB2, as well as TRPV1 receptors [35,40,47]. When CBD is applied to these cancer cells, it activates these receptors to induce late apoptotic programmed cell death via a caspase cascade [39]. 

The experimental groups of HeLa cells which received 0.125 µM ZnPcS_4_ PS plus 0.5 µM CBD without PDT reported a significant reduction in cell viability (35%**), and an even more significant 45%** of these cells were in a late apoptotic form of cell death. The significant cell death findings in the above results were very similar and comparable to the control groups of HeLa cells which received 0.5 µM CBD without irradiation. Since the control groups of HeLa cells which received 0.125 µM ZnPcS_4_ PS without PDT noted no significant forms of cell death due to the PS remaining inactive, this significant cell death could only be accredited to the treatment effects of CBD.

The experimental groups of HeLa cells which received 0.125 µM ZnPcS_4_ PS plus 0.5 µM CBD and PDT reported the most significant and favorable forms of cell death. Only 13%*** of the cells were viable and a significant 64%*** were in late forms of favorable apoptotic cell death. In comparison to the HeLa cells which received 0.125 µM ZnPcS_4_ PS plus 0.5 µM CBD without irradiation, the percentages of cellular viability were significantly decreased (35%** vs. 13%**) and the percentages of late apoptotic cell death were significantly increased (45%** vs. 64%***). Since the only difference between these two experimental groups was that irradiation was present, these findings suggested that the ZnPcS_4_ PS was successfully activated with laser irradiation and that this excitation was able to produce high yields of cytotoxic species, which promoted these substantial primary forms of late apoptotic cell death (64%***) in HeLa cells. 

Overall, these combinative experimental assays concluded that laser parameters at a wavelength of 673 nm and fluence of 10 J/cm^2^ did not affect this treatment pre-PDT. Furthermore, the application of laser light did not affect CBD’s ability to induce HeLa cell death. The ZnPcS_4_ PSs noted a high photochemical stability in the absence of light [15], and were capable of excitation in the presence of these laser parameters to produce high cytotoxic yields of ROS and ^1^O_2_ that could induce significant forms of late apoptotic cell death [34]. Both CBD and ZnPcS_4_ PSs were capable of significant amounts of programmed late apoptotic cell death, suggesting that this combinative treatment post-PDT would not induce any unwanted inflammatory responses or allow primary HeLa cells to recover [46].

### 3.5. WS1 Flow Cytometry Cell Death Pathway Quantitative Analysis 

The control groups of cells reported viable population of cells, with negligible cell death (Figure 6). These findings show that the laser parameters utilized in this study had no significant effects on WS1 normal human cell death when administered alone without the presence of the ZnPcS_4_ PS. These findings are supported by studies performed by Cios et al. (2021), which stated that red laser applications within the wavelength of 620 to 740 nm, without any additions of PS components, noted no difference in WS1 in-vitro-cultured biologic responses [48].

The control group of WS1 cells which received 99.8% (*v*/*v*) of ethanol only reported a majority of 88% of viable cells, without any other significant forms of cell death. Kar et al. (2021) stated that doses of ethanol lower than 10% (*v*/*v*) reported significant effects on in-vitro-cultured mouse fibroblasts cellular proliferation and viability; however, doses any higher than this, especially those utilized at clinical levels above 80% (*v*/*v*), remained non-toxic to the cells [49].

The control group of WS1 cells which received 0.125 µM ZnPcS_4_ PS alone, without any influence of laser irradiation, reported a noteworthy 85% of viable cells. Supportively, a study by Souza et al. (2021), which investigated the use of AlClPc as a possible photosensitizing agent for cancer PDT applications, noted no side effects in the normal in-vitro-cultured microenvironment of fibroblasts, in the absence of laser irradiation [50].

The control groups of WS1 cells which received 0.125 µM ZnPcS_4_ PS and PDT, did not report significant forms of cell death, suggesting that only slight cellular damage to normal human fibroblasts occurred. Insignificant early stages of apoptosis are often indicative of autophagy, whereby cells can sometimes recover [51,52]. Similarly, the subcellular localization assays noted that there was a slight uptake of the ZnPcS_4_ PS in the control normal cells, however it was far less when compared to the same control and experimental groups of HeLa cells. 

The control group of WS1 cells which received 0.5 µM CBD without irradiation noted 87% of cells being viable, whereas the experimental group which received 0.5 µM CBD with irradiation similarly noted 86% of cells being viable. Supportively, Pagano et al. (2020) reported that CBD doses ≤ 10 µM noted no cytotoxic effects on normal human healthy cell populations [53]. 

The experimental group of WS1 cells which received 0.125 µM ZnPcS_4_ PS plus 0.5 µM CBD with PDT noted no significant forms of cell death. However, in comparison to the cells-only control group, there was a higher population of early (10%) and late apoptosis (13%) being noted. Similar findings were reported in control groups which received 0.125 µM ZnPcS_4_ PS and PDT. Subsequently, control groups of WS1 cells which received 0.5 µM CBD with irradiation, respectively, showed slightly less (7%) early apoptosis and 4% of late apoptosis. Similarly, Souza et al. (2021) and Pagano et al. (2020) noted that neither red laser irradiation nor CBD caused unwanted cytotoxic effects in WS1 cells, respectively [50,53]. Avşar et al. (2016) noted that ZnPcS_2_ PSs showed a four-times higher intracellular uptake in the HeLa cells when compared to normal cell lines. The results concluded that even though there was slight absorption of the PS in normal cells, since laser irradiation was a localized tumor treatment, only some of the normal surrounding healthy tissues could become compromised [45]. 

### 3.6. HeLa ATP Quantitative Cell Proliferation Analysis

The findings substantiated previous discussions whereby it was noted that with irradiation alone, the CBD 99.8% (*v*/*v*) of ethanol solvent and the ZnPcS_4_ PS did not influence any combinative treatment outcomes in HeLa cells (Figure 7). However, the control groups of HeLa cells which received 0.125 µM ZnPcS_4_ PS and PDT, reported a significant decrease of 20%*** of the cellular population being able to proliferate post-PDT treatment. These results corroborated the findings in cell death analysis assays within the same control group, whereby an overall 48% of cell death (of which 15% of these cells were undergoing early apoptosis and a significant 24%** was late apoptotic) was found. As discussed above, Mishchenko et al. (2022) reported that the most favorable form of PDT-induced cell death is late apoptosis; however, if cancer cells undergo early apoptosis post-PDT, some could be in an autophagy state, which can simultaneously induce pro-survival and unwanted recovery [46]. Thus, these findings suggested that 0.125 µM ZnPcS_4_ PS and PDT treatment alone, is a feasible primary treatment for CC, however since it was unable to completely eradicate cells and a significant amount of the 20% of these cells remained in an active proliferative state, secondary spread could still occur. Even though research has reported that PDT has emerged as an effective and tolerable treatment strategy for the control of primary CC, it still requires improvement in relation to investigating enhanced PDT treatments capable of activating specific immune responses to fully eradicate metastasis [5].

The control and experimental group of HeLa cells which received 0.5 µM CBD, with or without irradiation, reported similar significant decreases in proliferation pre- and post-PDT treatment, respectively. These results corroborated the findings in cell death analysis assays within the same control group, which reported an insignificant number of cells undergoing early apoptosis on average, however noted a highly significant percentage of cells undergoing late apoptosis. The findings suggested that 0.5 µM CBD as a singular treatment of HeLa cells was capable of more significant forms of late apoptotic cell death when compared to 0.125 µM ZnPcS_4_ PS and PDT, which only induced a significant 24%** of late apoptotic cell death. This suggests that CBD can over-stimulate various cellular signaling pathways to generate late apoptotic and anti-tumor proliferation responses in CC cells [35].

The experimental groups of HeLa cells which received 0.125 µM ZnPcS_4_ PS plus 0.5 µM CBD without PDT reported a significant reduction, whereby only 18%*** of the cellular population were noted to have the abilities to proliferate. These results substantiate the findings in cell death analysis assays within the same control group, which reported on average an insignificant 11% of cells undergoing early apoptosis, however noted a highly significant 45%** of cells undergoing late apoptosis. The findings could only be attributed to the presence of CBD and not ZnPcS_4_ PS, as previously discussed, since it remained in an inactive state. Thus, these results further validated that CBD is capable of primary treatment CC, as well as generating anti-tumor immune responses capable of controlling metastatic tumor growth and spread [29].

Nevertheless, the experimental groups of the HeLa cells which received 0.125 µM ZnPcS_4_ PS plus 0.5 µM CBD and PDT reported the most significant decreases in cellular proliferation. These results confirm the findings in cell death analysis assays within the same control group, which reported an insignificant 7% of cells undergoing early apoptosis on average, however noted a staggering 64%*** of cells undergoing late apoptosis. These findings support previous statements by Lukhele and Motadi (2016), Ramer et al. (2010) and Contassot et al. (2004), whereby CBD was capable of significant forms of late apoptotic cell death in HeLa cells which are capable of anti-tumor proliferation responses, to prevent their metastasis in CC [35,40,47].

### 3.7. WS1 ATP Quantitative Cell Proliferation Analysis

The overall findings from this quantitative study noted that this combinative response, when activated with laser irradiation, was not capable of any significant forms of anti-proliferative effects in normal human WS1 fibroblast cells, when compared to the cells-only control (which received no forms of treatment) (Figure 8). These results safely implied that 0.125 µM ZnPcS_4_ PS plus 0.5 µM CBD with laser irradiation should not affect the biological responses of normal human tissues when applied in combination for the treatment of CC post-PDT. Furthermore, these results suggest that the minor early apoptotic forms of cell death, which were reported in the cell death assays in normal WS1 fibroblasts caused no unwanted cytotoxic effects and supported previous discussions, stating that these incremental amounts of cells were probably in an autophagic state and were able to recover post-PDT. Nevertheless, even these insignificant forms of cellular damage could be overcome in future studies, and the overall PDT treatment outcomes of this study could be improved upon if this ZnPcS_4_ PS was conjugated to an active targeting nanocarrier, to ensure that its cellular absorption is only taken up by cancer cells [34]. 

## 4. Materials and Methods

### 4.1. Cell Culture 

Commercially purchased CC cell line HeLa (Cellonex CAT SS1411) was used in this study as the CC-positive cell line, whereas WS1 human skin fibroblasts procured from the American Type Culture Collection (ATTC CRL-1502) were used for result comparison outcomes to represent normal human tissues. Cells were cultivated in a T25 culture flask containing supplemented pre-warmed media. 

HeLa cells were cultivated in Dulbecco’s modified eagle medium (DMEM), enriched with 10% fetal bovine serum (FBS), 4 mM sodium pyruvate, 4 mM L-glutamine, 2.5 g/mL amphotericin β, and 100 U penicillin 100 g/mL streptomycin solution. WS1 cells were cultivated in minimum essential media (EMEM), enriched with 2 mM L-glutamine, 1 mM sodium pyruvate, 0.1 mM nonessential amino acids, 10% FBS, 2.5 g/mL amphotericin β and 100 U penicillin 100 g/mL streptomycin solution. These re-constituted cell cultures were then incubated at 37 °C in 5% CO_2_ and 85% humidity. Once 90% confluent monolayers of the cells were detached from the flasks using TrypLE Select™, consequent cellular pellets were re-suspended in fresh culture media.

Thereafter, HeLa cells were seeded at a density of 5 × 10^5^ cells/mL and WS1 cells at 3 × 10^5^ cells/mL of supplemented media into sterile 3.4 cm diameter cell culture plates. The plates were incubated for 4 h to allow for cellular attachment, prior to conducting experiments in the sterile tissue-culture-treated plates. 

### 4.2. Cell-Culture Plate Groupings and Laser Irradiation

After 4 h, HeLa and WS1 culture plates were divided into the various control and experimental groups for CBD or ZnPcS_4_ PS PDT individual dose response assays, and combinative experiments. 

The cell- and irradiation-only control groups had their media simply replaced with fresh media. The control or experimental groups which needed varying or concentrations of ZnPcS_4_ PS or CBD alone, had these concentrations added to their fresh media. All culture plates were then re-incubated for an additional 20 h. Then, the PDT experimental and control groups were irradiated in the dark using a Roithner 1000 mA 673 nm high-power semiconductor diode laser, at a fluency of 10 J/cm^2^ for average time of 16 min and 8 sec [16]. After irradiation the old cell culture media from all culture plates was removed and replaced, following re-incubation of all culture plates for an additional 24 h before biochemical assessments.

### 4.3. Chemicals

The zinc (II) phthalocyanine tetrasulfonic acid (ZnPcS_4_) PS used in this study was purchased from Santa Cruz^®^ Biotechnology (Dallas, TX, USA) (sc-264509A, molecular weight 898.15 g/mol) (Figure 9a). ZnPcS_4_ powder was solubilized in 0.001 M phosphate buffered saline (PBS). It exhibits three major Q-bands at emissions of 583, 634, and 674 nm, within the far-red spectral range, considered ideal for primary PDT cancer treatment [39]. To make a working stock concentration of 125 µM, 1 mL of 0.0005 M ZnPcS_4_ PS stock solution was diluted with 4 mL of 0.001 M PBS. This working stock concentration of ZnPcS_4_ PS was then further diluted in complete cell culture media, to acquire varying dose concentrations (0.0625, 0.125, 0.25, 0.5, and 1 µM), which were utilized within the below discussed assays [34].

CBD was administered to CC cells pre-PDT treatment. The 10 mg/mL CBD solution solubilized in 1 mL 99.8% ethanol, with a molecular mass of 314.46 g/mol, was commercially obtained from Sigma-Aldrich (St. Louis, MO, USA) (90899, 1 mL) (Figure 9b). The CBD was diluted with 19 mL of 99.8% of ethanol to make a stock concentration of 0.5 mg/mL. CBD working stock solution was stored in the dark in a refrigerator. This working stock concentration of CBD was then further diluted in complete cell culture media to acquire varying dose concentrations (0.3, 0.5, 0.7, 0.9, and 1.1 µM), which were utilized within the below-discussed assays [35].

### 4.4. Individual ZnPcS_4_ PS PDT or CBD Irradiation LDH Cytoxicity Dose Response Assays

Individual dose response studies were performed using the inhibitory concentration dose (ICD) method to determine the lowest concentration dose of ZnPcS_4_ PS or CBD alone that could yield 50% cytotoxicity (ICD_50_), so that cellular viability could be ensured post-experimentation within combinative experimentation, to allow for biological result interpretations. 

Following 4 h incubation, HeLa or WS1 cell control or experimental culture plate groups which required ZnPcS_4_ PS in varying doses received it in its diluted form through their culture media. Four hours post incubation, control or experimental culture plate groups which required CBD varying doses or ICD_50_ concentrations received it in its diluted 99.8% ethanol form through their culture media. The culture plates were then re-incubated in the dark for an additional 20 h, after which the control and experimental groups which required irradiation received it. All culture plates had their media replaced and they were re-incubated again for 24 h before conducting LDH dose response assays. 

The CytoTox 96^®^ non-radioactive cytotoxicity assay kit (Promega™ G1780, Madison, WI, USA) was used to quantitatively measure lactate dehydrogenase (LDH), which is a cytosolic enzyme that is released upon cell lysis, that is directly proportional to cellular cytotoxicity. Briefly, 24 h post PDT, 50 μL of complete cell culture media supernatant from each experimental and control culture plate was removed and mixed with 50 μL of LDH reconstituted substrate mix in flat bottom 96-well microplates. To determine cytotoxicity from the LDH cellular lysis that was produced post treatment, LDH absorbance was measured at 490 nm, using a spectrophotometer (Perkin Elmer, Victor3, 1420 Multilabel Counter, Waltham, MA, USA).

The above dose response assays reported that the ICD_50_ post PDT for ZnPcS_4_ PS and CBD was found to be 0.125 µM and 0.5 µM, respectively. Thus, 24 h post treatment, both of these concentrations were utilized in the below-combinative subcellular localization, flow cytometry cell death, and cellular proliferation assays, to determine the combined effects that ZnPcS_4_ PS and CBD had on CC and WS1 post treatment.

### 4.5. Quantitative Subcellular Localization Immunofluorescent Staining Confirmation of ZnPcS_4_ PS Uptake 

HeLa cells or WS1 cells were detached from cell culture flasks and seeded in culture plates, which had sterile cover slips inserted into their base at previously mentioned densities. After 4 h incubation to allow for cellular attachment of the coverslips, the complete growth medium from the culture plates was replaced. The culture plates were then divided into various experimental and control groups, and received varying volumes of complete cell culture media and pre-determined ICD_50_ 0.125 µM ZnPcS_4_ PS and/or 0.5 µM CBD to accommodate for the required combinative PDT assays when both were applied in combination. The culture plates were then re-incubated in the dark for an additional 20 h. 

All experimentation was conducted in the dark, to ensure that the PS or fluorochromes used did not photo-bleach. Following incubation, the complete culture media in all culture plates were discarded, then placed on ice and rinsed with 1 mL ice-cold 0.01 M PBS. Then, 1 mL of 3.7% (*v*/*v*) formaldehyde solution was added for 10 min to allow for cellular fixation to the coverslips. Thereafter, 1 mL of rinsing solution consisting of ice-cold 0.01 M PBS in 1% (*w*/*v*) bovine serum albumin (BSA) and 0.01% (*w*/*v*) sodium azide buffer solution was briefly added and then discarded. The cells were then stained with 200 µL diluted primary antibody (2 µg/mL ICAM-1 mouse monoclonal IgG1: AbAB2213 AC Abcam) for 30 min on ice and washed with rinsing solution. Then, the cells were stained with 200 µL diluted secondary antibody (5 µg/mL goat anti-mouse IgG-FITC: AB6785 AC Abcam) for 30 min on ice and washed with rinsing solution. Next, 50 µL of 1 µg/mL 40-6-Diamidino-2-phenylindole (DAPI) was added, and the culture plates were incubated for 5 min at room temperature, after which they were rinsed with 1 mL HBSS. The coverslips were then removed from the culture plates and attached to glass microscope slides with 50 µL of mounting medium. The slides were examined using the filter settings of a Carl Zeiss Axio Z1 Observer immunofluorescent microscope at 40× magnification. 

The 358Ex/461Em filter was used to detect blue-DAPI-counter-stained nuclei in cultured cells, while the 495Ex/519Em filter was used to detect any green-FITC-stained ICAM-1 membrane proteins in cultured cells. ICAM-1 proteins are cellular surface membrane markers that are expressed within almost all in-vitro-adherent cells [54]. Thus, ICAM-1 proteins may be used to indirectly (when conjugated to a fluorochrome, such as FITC) counter-stain in-vitro-cultured cells membranes, to allow for identification and subcellular location of internal cellular contents when conducting immunofluorescent microscopy experiments. Lastly, the 589Ex/610Em filter was used to detect any Cy5 red fluorescent signals that were produced by the ZnPcS_4_ PS within the various control and experimental groups, to determine if the PS was capable of subcellular localization in HeLa cells only, and whether it had no effective uptake in WS1 cells when combined with CBD.

### 4.6. Flow Cytometry Cell Death Pathway Analysis of ZnPcS_4_ PS and CBD PDT/Irradiation Combinative Assays

The Annexin V-FITC/PI cell death detection kit (BD Pharmingen™ Scientific: BD/556570) was used for the detection and quantitation of cells undergoing early or late apoptosis, cells dying from necrosis cells, or cells remaining viable within combinative ZnPcS_4_ PS and CBD PDT response assays 24 h post irradiation, using a BD Accuri™ C6 flow cytometer. All experimentation instructions and controls for gating were included as per the manufacturers’ protocol recommendations.

Briefly, with reference to Appendix A cellular suspensions were centrifuged in microcentrifuge tubes at 2200 rpm for 4 min at 20 °C. Their supernatants and cell pellets were re-suspended in 1× (*v*/*v*) binding buffer to a 1 × 10^5^ cells/mL. The samples were centrifuged, and 100 µL of each cellular suspension was stained with 5 µL of Annexin V-FITC solution and 5 µL of reconstituted propidium iodide staining solution. Before the acquisition, 400 μL of binding buffer 1× was added. For each sample, 20,000 events were acquired using a BD Accuri™ C6 flow cytometer, with a 488 nm solid-state laser (40 mW), and optimal photomultiplier (PMT) voltages were established for each channel. The matching BD Accuri C6 Plus Annexin V-FITC/PI software kit template was used for data acquisition. Debris and doublets were excluded from the analysis. Early and late apoptotic cells were identified for their positivity to Annexin V, and necrotic cells were identified for their positivity to PI, as well as live cells being identified as per non-stain controls, as shown in the FACS gating strategy represented in Appendix A. The FlowJo v 10.8.1 Software (BD Biosciences) was used for data analysis. To obtain comparable results, prior to sample analyses, all quality control procedures and calibrations were performed on the flow cytometer using BD Pharmingen™ Scientific Flow Maintenance and Flow Starter Kits, with tracking beads. 

### 4.7. Adenosine Triphosphate (ATP) Cell Proliferation Analysis of ZnPcS_4_ PS and CBD PDT/Irradiation Combinative Assays

The CellTiter-Glo^®^ luminescent cell viability assay (Promega™ PRG7571) was used to determine the number of metabolically active cells which can generate adenosine triphosphate (ATP) post irradiation. The assay utilizes the properties of a commercial thermostable luciferase, which generates a luminescent signal proportional to the amount of ATP released upon cell lysis, and so signifies their overall viability and abilities to recover/proliferate post treatment. 

The cellular proliferation of HeLa and WS1 cells post-irradiation for the various control and experimental groups of combinative ZnPcS_4_ PS and CBD PDT response assays was measured to determine the combinative anti-proliferative effects this treatment had on HeLa, as well as determine if there were any unwanted side effects on normal cells.

Briefly, 100 μL of reconstituted reagent was added to an equivalent volume of cell culture plates cell suspension post treatment. The contents were then pipetted into a 96-well white-walled microplate, and stirred together on a shaker for 2 min to induce cell lysis. The microplate was then incubated at room temperature for 10 min to allow for the luminescent signal to stabilize. The luminescent signal was determined in relative light units (RLUs) using a Perkin Elmer, Victor^3^, multilabel high-performance multiplate counter (model 1420). The amount of ATP detected was quantified in direct proportion to the number of proliferating cells post treatment.

### 4.8. Statistical Analysis

Experimental assays were all performed in triplicate and repeated over three different days, and an average of these results was reported. Raw data were captured using Microsoft excel sheets and statistical analysis was done using the GraphPad Prism version 9 software to determine the mean, standard deviation (SD), standard error, and significant changes for the data when comparing the various control and experimental groups. The Student *t*-test and one-way analysis of variance (ANOVA) were used for comparing two or more means of normally distributed data, whereas the Mann–Whitney test was used for non-normally distributed data to determine the statistical difference between the various control and experimental groups. The error bars on graphs display the mean standard error (SEM), and if an experimental or control group reported a SD higher than 1 within a sample repeat, it was repeated to ensure data accuracy and precise statistical comparisons. Statistical differences between the various control and experimental groups were determined. Values within the 95% confidence interval (*p* < 0.05*, *p* < 0.01** or *p* < 0.001***) were accepted as statistically different and represented graphically.

## 5. Conclusions and Future Research Direction 

### 5.1. Conclusions

The findings of this study presented sufficient evidence to quantitively observe the in vitro subcellular localization efficiency of ZnPcS_4_ PS, as well as determine that ZnPcS_4_ PS and CBD in cultured HeLa CC cells were cable of inducing late apoptotic forms of cell death, proposing that this combinative treatment approach could possibly effectively allow PDT to annihilate primary CC tumors, with CBD secondary tumor inhibition, as well as beneficially induce nominal effects on healthy cells. 

### 5.2. Future Research Direction

In relation to trying to address some of the shortfalls noted in PDT cancer therapeutic treatments, future research into this combinative ZnPcS_4_ PS PDT and CBD treatment would require in vitro research in enhancing the uptake of PS and CBD via active targeting nanosystems [28]. This may potentially circumvent any unwanted uptake in normal cells, as well as reduce photosensitivity. Furthermore, studies by Thakur et al. (2019) developed a distinct photo-amenable NP-based drug delivery system, which facilitated efficient targeted on-demand delivery, fluorescence imaging, and therapy, by incorporating a ZnPc PS and gold NPs into liposomes to try overcome its hydrophobic nature [55]. These in vitro experiments exhibited more than 85% cytotoxicity in MCF-7 breast cancer cells [55]. In vivo studies in tumor-bearing Sprague Dawley rats showed that the photodynamic anti-cancer activity of ZnPc PS gold NP liposomes with a quercetin flavonoid was increased at the tumor site with minimal systemic toxicity [56].

In addition, numerous studies have noted that the failure of PDT within in vivo and clinical applications is due to the hypoxic environment that solid tumors possess. This limits the amount of molecular oxygen available in a tumor’s microenvironment, when a PS is activated to produce sufficient ROS and ^1^O_2_ to effectively kill tumor cells [19]. However, recent advances in nanotechnology have allowed for the development of anti-hypoxia nanotechnology-mediated PS delivery platforms, which can address these issues by increasing oxygen supply, allowing for the disruption of a tumor’s extracellular matrix (ECM), inhibition of O_2_ consumption and other angiogenic factors [19,57]. Zhu et al. (2020) investigated the combination of PDT with enzyme therapy for malignant tumor therapies, as it takes advantage of the spatial-controlled PDT and the effective enzyme-catalyzed bioreactions, to create a synergistic starvation therapy and PDT [58]. The study developed a versatile strategy for delivering hydrophilic enzymes and a hydrophobic PS using protocell-like nanoreactor (GOx-MSN@MnPc-LP) [58]. After uptake, the nanoreactor depleted endogenic glucose in tumor cells, and so generated considerable toxic H_2_O_2_ by GOx, as well as produced intracellular ROS under laser irradiation by MnPc [58]. The antitumor effects of the nanoreactor were verified on tumor cells and tumor-bearing mouse models, and reported that it efficiently inhibited tumor growth in vivo with a single treatment [58].

Clinically, most treatments used for cancers induce higher expressions of programmed cell death ligand 1 (PD-L1), which impairs the efficacy of anti-cancer drugs [59], including PDT. Moreover, it has recently been discovered that the cytoplasm or nucleus-distributed PD-L1 protein may also detrimentally limit the efficacy of certain therapies by increasing DNA damage repair [60]. Interestingly, recent experimentation has revealed that certain PDT PSs, when coupled with appropriate anti-tumor agents, can potentially overcome the challenges of PD-L1 up-regulation and tumor hypoxia in PDT, and so ultimately improve treatment outcomes [60]. In a study by Zhou et al. (2022), the PS drug carriers, Butformin (Bu) and methylene blue (MB), that possess two-stage oxygen delivery capacity and PD-L1 cascade inhibition ability, were effectively loaded into MB@Bu@MnO_2_ nanoparticles. Due to the enhanced oxygen delivery of MnO_2_ (manganese dioxide)-mediated oxygen generation and reduced oxygen consumption induced by Bu-mediated mitochondrial dysfunction, the hypoxia tumor microenvironment was reversed, leading to enhanced ROS generation [59]. The selectively released Bu also reduced the PD-L1 protein expression in tumor cells (both in vivo and in vitro) via enhanced adenosine 5′-monophosphate-activated protein kinase (AMPK) phosphorylation, resulting in significant increases of CD8+, CD4+, and CD3+ T-cell infiltration in vivo. Finally, the reversed tumor hypoxia mediated by two-stage oxygen delivery and the enhanced T-cell infiltration induced by PD-1/PD-L1 axis cascade inhibition helped overcome limitations of conventional PDT, improving its clinical efficacy [59].

In another study, researchers investigated the cascade two-stage tumor re-oxygenation and immune re-sensitization mediated by self-assembled albumin-sorafenib NPs for enhanced photodynamic immunotherapy [57]. They designed cascade two-stage re-oxygenation and immune re-sensitization BSA-MHI148@SRF NP, with a near-infrared photodynamic dye MHI148, chemically modified with bovine serum albumin (BSA) and multi-kinase inhibitor Sorafenib (SRF), to serve as a novel tumor oxygen and immune microenvironment regulation drug [57]. The SRF decreased tumor oxygen consumption by inhibiting mitochondria respiration, and increased the oxygen supply via inducing tumor vessel normalization [57]. The enhanced immunogenic cell death production was amplified by BSA-MHI148@SRF to induce PDT ROS generation, as well as enhance T-cell infiltration, and so the tumor cell killing ability was improved [57]. The BSA-MHI148@SRF amplified tumor vessel normalization via VEGF inhibition and thus reversed the tumor immune suppression microenvironment [57]. Overall, the growth of solid tumors in murine cancer models was significantly depressed, and this nano re-oxygenation and immune re-sensitization system could therefore potentially promote PDT to clinical trials [57].

When considering the fundamentals of this study, there is still far more in vitro research required to bring this ZnPcS_4_ PS PDT and CBD combinative treatment approach to possible in vivo and clinical translations. However, it does raise hope for CC treatment. 

## Figures and Tables

**Figure 1 ijms-24-06151-f001:**
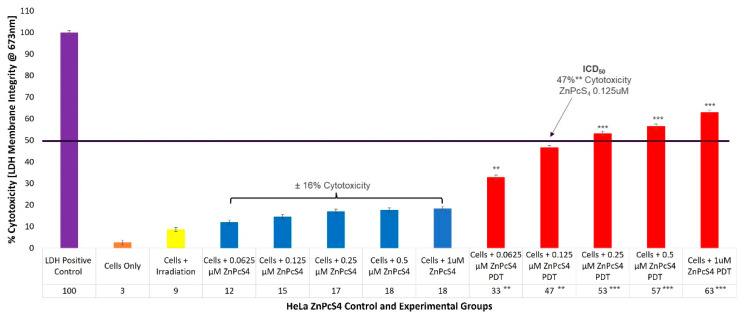
LDH membrane integrity response of HeLa cells treated with varying increasing concentrations of ZnPcS_4_ PS, demonstrating a significant dose-dependent increase in cellular cytotoxicity post PDT (*p* < 0.01** or *p* < 0.001***).

**Figure 2 ijms-24-06151-f002:**
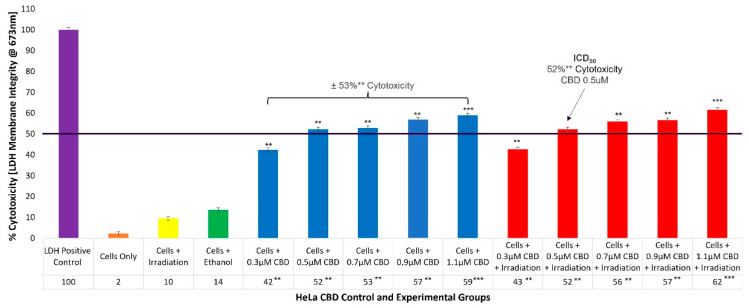
LDH integrity response of HeLa cells treated with varying increasing concentrations of CBD, demonstrating a significant dose-dependent increase in cellular cytotoxicity pre- and post-irradiation (*p* < 0.01** or *p* < 0.001***).

**Figure 3 ijms-24-06151-f003:**
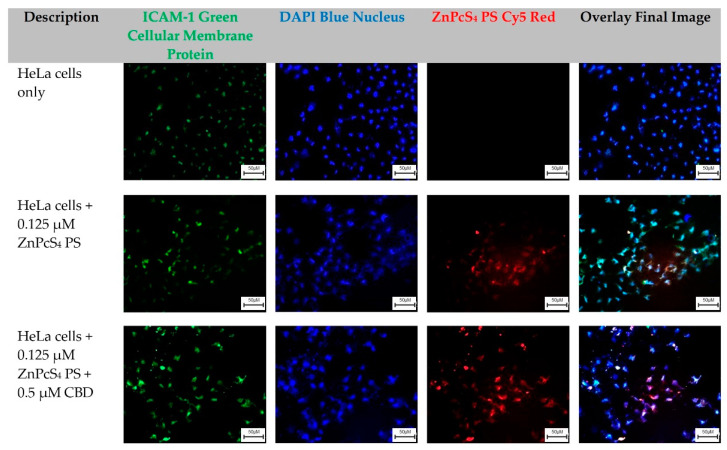
Subcellular localization comparison of various control and experimental groups of HeLa cells treated with singular and combinative ZnPcS_4_ PS and/or CBD pre-irradiation; DAPI-stained nuclei (blue), ICAM-1 cellular membrane proteins (green) and Cy5 fluorescence from ZnPcS_4_ PS subcellular localization (red) (40× magnification, 50 µM scale bar).

**Figure 4 ijms-24-06151-f004:**
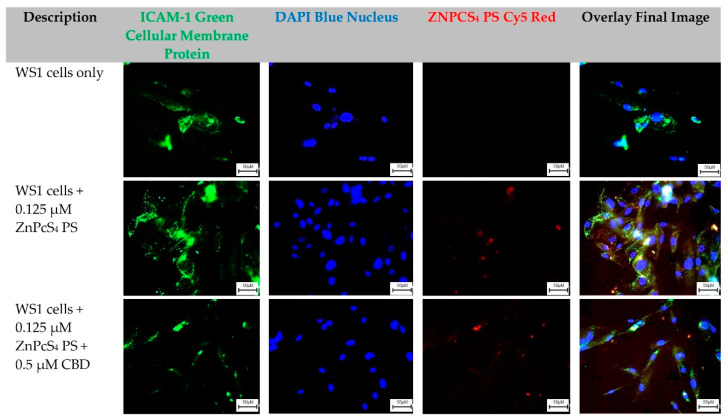
Subcellular localization comparison of various control and experimental groups of WS1 normal fibroblast cells, treated with singular and combinative ZnPcS_4_ PS and/or CBD pre-irradiation; DAPI-stained nuclei (blue), ICAM-1 cellular membrane proteins (green), and Cy5 fluorescence from ZnPcS_4_ PS subcellular localization (red) (40× magnification, 50 µM scale bar).

**Figure 5 ijms-24-06151-f005:**
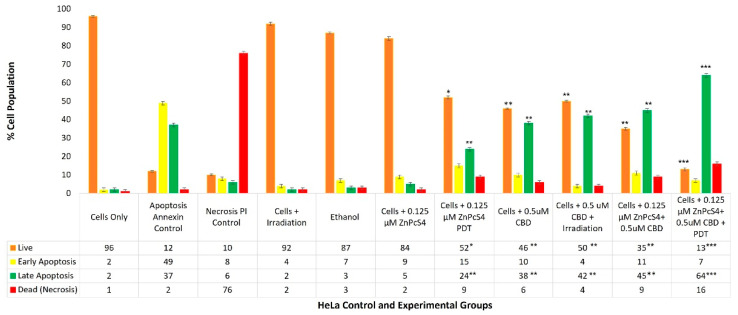
Percentage of different stages of cell death using the flow cytometry Annexin V-FITC/PI staining method on various HeLa control and experimental groups, within ZnPcS_4_ PS and CBD PDT/irradiation combinative assays (*p* < 0.05*, *p* < 0.01** or *p* < 0.001***).

**Figure 6 ijms-24-06151-f006:**
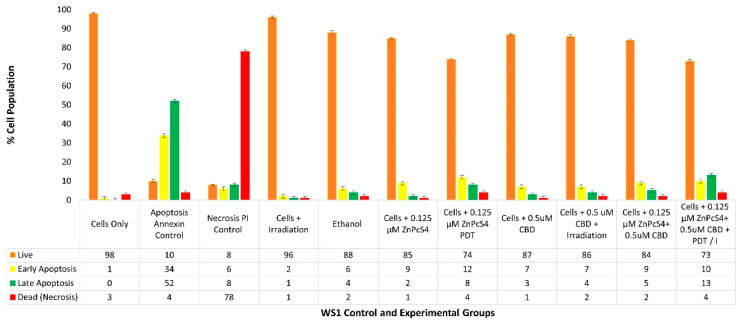
Percentage of different stages of cell death using the flow cytometry Annexin V-FITC/PI staining method on various WS1 normal fibroblast control and experimental groups, within ZnPcS_4_ PS and CBD PDT/irradiation combinative assays.

**Figure 7 ijms-24-06151-f007:**
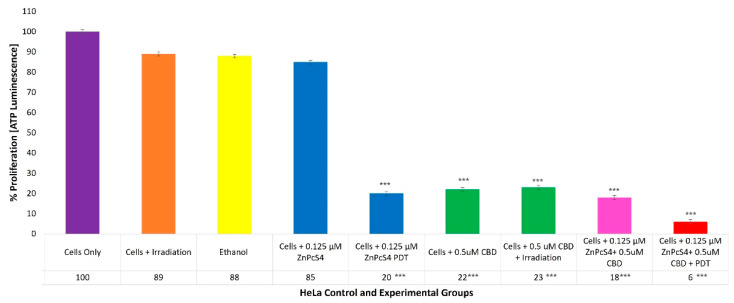
Percentage of cell proliferation among various HeLa control and experimental groups, within ZnPcS_4_ PS and CBD post-irradiation combinative assays (*p* < 0.001***).

**Figure 8 ijms-24-06151-f008:**
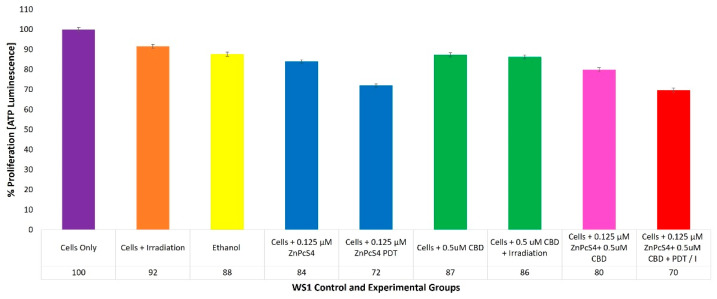
Percentage of cell proliferation of various WS1 normal fibroblast control and experimental groups, within ZnPcS_4_ PS and CBD post-irradiation combinative assays.

**Figure 9 ijms-24-06151-f009:**
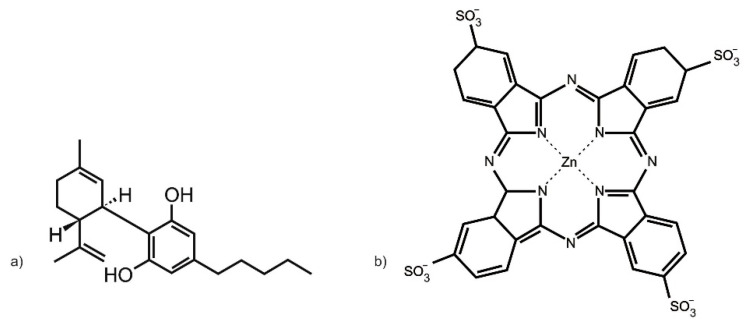
(**a**) Chemical structure of CBD. (**b**) Chemical structure of ZnPcS_4_ PS.

## Data Availability

The data presented in this study are available on request from the corresponding author. The data are not publicly available due to the privacy of unpublished data sets.

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
