# Peer review of "Cytotoxic Effects of Combinative ZnPcS_4_ Photosensitizer Photodynamic Therapy (PDT) and Cannabidiol (CBD) on a Cervical Cancer Cell Line"

_ijms, 2023, doi:10.3390/ijms24076151_

Round 1

Reviewer 1 Report

In this research, the authors evaluated the cytotoxic effects of combinative photodynamic therapy (PDT) and Cannabidiol (CBD) on cervical cancer cells. In my opinion, the current version of this manuscript fits the scope of International Journal of Molecular Sciences and could be accepted after minor revision.

My specific comments are in detail listed below:

1.     The columns of Figure 1 and Figure 2 is of low quality. The authors should further modify or improve it.

2.     All the data mentioned should be presented as mean ± SD rather than the only the average.

3.     The introduction still could be improved, especially the current development of PDT in the Line 44-52, some references may be helpful to the authors, including
doi.org/10.1002/adma.202206121.

4.     It’s confusing to show Table 1-2 as Tables. In my opinion, it’s better to draw pictures to present these data.

5.     In my opinion, the advantage and shortcomings of using Cannabidiol to enhance the efficacy of PDT should be more clearly discussed in the results or discussion part. Some related references with the relevant development of PDT sensitizers should be added including

1.doi.org/10.1016/j.jconrel.2022.11.004  
2.doi.org/10.1016/j.apsb.2022.07.023
The quality of Figure 3-6 should be proved to better show the data.

6.     In all the tables or data shown in the results, all the data shown are with no decimal place. It’s better to reserve one or two decimal fraction.  

Author Response

REVIEWER 1

Thank you for your feedback in enhancing the quality of our manuscript submission. These have been addressed by the authors, per reviewer’s comment, and responded to below in red. All requested corrections have been indicated in red font in the revised manuscript. In addition to noting changes in reference numbers, due to inserts and figure number changes.

Comments and Suggestions for Authors

In my opinion, the current version of this manuscript fits the scope of International Journal of Molecular Sciences and could be accepted after minor revision.

My specific comments are in detail listed below:

  1. The columns of Figure 1 and Figure 2 is of low quality. The authors should further modify or improve it.

Corrected; the quality of all the figures have been enhanced, and 600DPI images have been supplied.

  1. All the data mentioned should be presented as mean ± SD rather than the only the average.

Corrected; the accepted ±SD has been mentioned in Section 4.8. Statistical Analysis and the average standard error mean ± SEM, has been included throughout when average percentage results have been reported.

  1. The introduction still could be improved, especially the current development of PDT in the Line 44-52, some references may be helpful to the authors, including

doi.org/10.1002/adma.202206121.

Thank you… this useful reference has been included and elaborated upon in the PDT section.

  1. It’s confusing to show Table 1-2 as Tables. In my opinion, it’s better to draw pictures to present these data.

Corrected; both Table 1 and 2 have been corrected to Figures 3 and 4, and labelling been changed to figure legends.

  1. In my opinion, the advantage and shortcomings of using Cannabidiol to enhance the efficacy of PDT should be more clearly discussed in the results or discussion part. Some related references with the relevant development of PDT sensitizers should be added including:

1.doi.org/10.1016/j.jconrel.2022.11.004 

2.doi.org/10.1016/j.apsb.2022.07.023

The aforementioned studies by Zhou et al., (2022) related to “Metabolic reprogramming mediated PD-L1 depression and hypoxia reversion to reactivate tumor therapy” and “Cascade two-stage tumor re-oxygenation and immune re-sensitization mediated by self-assembled albumin-sorafenib nanoparticles for enhanced photodynamic immunotherapy” have been included in the conclusion part of this manuscript.

  1. The quality of Figure 3-6 should be proved to better show the data.

Corrected; the quality of all the figures have been enhanced, and 600DPI images have been supplied.

  1. In all the tables or data shown in the results, all the data shown are with no decimal place. It’s better to reserve one or two decimal fraction.

Decimal places were rounded up to whole numbers making the manuscript easier to read and reduce the burden on readers. This round up to whole numbers had no impact on the representation of the figures, results reporting or data analysis. Raw data will be provided should the reader wish to access it from the for the noted raw data repository link in the data availability statement once published.

Reviewer 2 Report

This article is devoted to the study of combined cytotoxic effect of photodynamic treatment with zinc tetrasulfonated phthalocyanine (ZnPcS4) and cannabidiol. Authors demonstrate that: (1) separately cannabidiol or PDT with ZnPcS4 are not able to kill more than 60% cancer HeLa cells in vitro, but they can kill up to 90% of cells if applied together; (2) as “mono”, as combined treatments provide cell death via activation of apoptosis; sensitivity of WS1 fibroblast to the combined treatment is noticeably lower than that of HeLa cells.    

According to my opinion, this article demonstrates a rather trivial result: cytotoxicity of inefficient photosensitizer (PS) can be improved in vitro if it is combined with another cytotoxic agent.

It is unclear, if this effect can be reproduced in vivo, and whether it will provide significant anticancer outcome. Accordingly, all speculations (see, for example, abstract lines 30-33) about possible advantages of such combined treatment for anticancer therapy are superficial.         

According to my experience in PDT research, PSs that are unable to kill 100% cancer cells in vitro are inefficient PS for anticancer PDT in vivo. Indeed, ZnPcS4 is inefficient anticancer photosensitizer as it was already verified in vivo by several research groups twenty years ago during search for clinically promising photosensitizers. If you disagree, please find and cite papers, where ZnPcS4 provided essential decrease in tumor growth or its eradication. Any statements on efficacy of PS for anticancer PDT that are solely based on its ability to kill a part of cancer cells in vitro are erroneous.

The authors of the article either do not know the subject of research, or deliberately distort it in order to enhance the significance of experiments with ZnPcS4 (see lines 62-67).  

I'll explain. Tetra-sulfonated phthalocyanines (no matter metalless, with zinc or aluminum) penetrate very weakly into cells due to a large negative charge and therefore have a low ability to photodynamically damage cancer cells. It is necessary to reduce the number of sulfogroups in phthalocyanine in order to enhance its penetration into cancer cells and its photocytotoxicity. Phthalocyanine with a number of sulfogroups less than 4 can be obtained only as a mixture of variously sulfonated derivatives: mono, di, tri, tetra. Therefore, researchers are trying to find the optimal balance and obtain a mixture of phthalocyanines enriched with the most active disulfonated derivatives, while maintaining the solubility of the mixture of phthalocyanines.

Taking into account this reality, authors should not cite the data about cytotoxicity or antitumor activity of low sulfonated phthalocyanines as a justification for selection of ZnPcS4. They should not speculate that improved solubility of ZnPcS4 is advantage for anticancer PDT.

I see that the authors violate the traditional ethics of scientific publications by manipulating statements and references allegedly confirming them. Below are two examples.

Lines 50-58. Incorrect and unsupported statement: authors of [9] did not study either “triplet ROS yields” or “ maximum  penetration of light into the tissues” for Pc PSs.  

Lines 64-67. Incorrect and unsupported statement: authors of [11] did not perform any studies proving that  “four sulfhydryl/thiol acetate groups … enhance PS solubility and so promote enhanced tumor subcellular localization and overall PDT treatment outcomes”.

It seems that all similar statements and their support by references should be carefully checked, but it is not a duty of reviewer, it is the duty of authors.  

Among other drawbacks of the manuscript are artificial results presented in so called Table 1 and partially in Table 2 (in fact, they are Figures, not Tables). Why do the membrane probe ICAM-1 and nuclear probe DAPI have similar distribution in cells (see especially Table 1)? Why do ZnPcS4 and nuclear probe DAPI have similar distribution in many cells, if it is known that  ZnPcS4 accumulates in cytoplasm not in the nucleus?

Another major drawback is unreasonably long Discussion section containing extensive repetitions of data and statements from Introduction and Results sections.                 

Apart of some typos, terminology used by authors, many statements and even figure legends require careful verification and edition. As examples see legends to figures 3-6, (line 366) “during subcellular tumor absorption” and (line 874) “Cy5 red auto fluorescent signals “.

Section Materials and methods contains trivial descriptions, like (lines 798-799) to “make a stock concentration of 0.0005 M, 0.0006 g of ZnPcS4 powder was added to 1.25 ml of 0.001 M PBS” and does not contain essential details for flow cytometry measurements, parameters and statistics.    

Finally, I conclude that the presented manuscript is not ready for publication in any scientific journal and especially in highly rated journals like IJMS.              

Author Response

REVIEWER 2

Thank you for your feedback in enhancing the quality of our manuscript submission. These have been addressed by the authors, per reviewer’s comment, and responded to below in red. All requested corrections have been indicated in red font in the revised manuscript. In addition to noting changes in reference numbers, due to inserts and figure number changes and a full English editorial has been done.

Comments and Suggestions for Authors

This article is devoted to the study of combined cytotoxic effect of photodynamic treatment with zinc tetrasulfonated phthalocyanine (ZnPcS4) and cannabidiol. Authors demonstrate that: (1) separately cannabidiol or PDT with ZnPcS4 are not able to kill more than 60% cancer HeLa cells in vitro, but they can kill up to 90% of cells if applied together; (2) as “mono”, as combined treatments provide cell death via activation of apoptosis; sensitivity of WS1 fibroblast to the combined treatment is noticeably lower than that of HeLa cells.   

  • According to my opinion, this article demonstrates a rather trivial result: cytotoxicity of inefficient photosensitizer (PS) can be improved in vitro if it is combined with another cytotoxic agent.

The authors disagree. ZnPcS4 has been demonstrated as an efficient PS in more recent studies. In addition, Pcs present high photo and chemical stability. The authors however agree that more efficient ways to enhance this PS delivery and uptake in cells should be investigated in future studies to enhance the potential of effective treatment in cancers with PDT, which have been elaborated upon in the manuscript. Furthermore, combinative PDT treatment approaches have also shown superior efficacy.

  • According to my opinion, this article demonstrates a rather trivial result: cytotoxicity of inefficient photosensitizer (PS) can be improved in vitro if it is combined with another cytotoxic agent.

The authors disagree. This article demonstrates a rather efficient photosensitizer (PS) used in combination with CBD demonstrates significant cytotoxic activity and cell death in in vitro cervical cancer, yet preserves the integrity of healthy cells, and warrants further research.

  • It is unclear, if this effect can be reproduced in vivo, and whether it will provide significant anticancer outcome. Accordingly, all speculations (see, for example, abstract lines 30-33) about possible advantages of such combined treatment for anticancer therapy are superficial.

As this is novel in vitro research, further in vitro and in vivo studies will help to address this question. Nevertheless, PDT has made great progress in clinical practice due to its high selectivity and low side effects. However, there is still a lot of work to do to improve PDT in its penetrability to tumor tissue, selectivity, stimulating methods and promote its ability to overcome tumor hypoxia microenvironment. This has been addressed in detail within the introduction the Conclusion and Future Research Direction part of this manuscript.

  • According to my experience in PDT research, PSs that are unable to kill 100% cancer cells in vitro are inefficient PS for anticancer PDT in vivo. Indeed, ZnPcS4 is inefficient anticancer photosensitizer as it was already verified in vivo by several research groups twenty years ago during search for clinically promising photosensitizers. If you disagree, please find and cite papers, where ZnPcS4 provided essential decrease in tumor growth or its eradication. Any statements on efficacy of PS for anticancer PDT that are solely based on its ability to kill a part of cancer cells in vitro are erroneous.

The below recent published articles have been included throughout this manuscript which make reference to the excellent research in vitro outcomes utilizing ZnPcS4 PS as a PDT PS for cancer:

  • Pola et al. (2020) investigated the PDT effects of a disulphonated zinc phthalocyanine (ZnPcS2) PS within in vitro HeLa cells and results noted that the PS was capable of subcellular localization and that this PDT inhibited mitochondrial respiration in hypoxic conditions.
  • In vitro studies performed by Chekwube et al. (2020) successfully investigated the phototoxic effectiveness of ZnPcS4 PS in MCF-7 breast cancer cells, with significant cytotoxic effects being reported.
  • Studies by de Toldeo et al. (2020) demonstrated the improved uptake of ZnPcS4 PS when it was encapsulated in poly (lactic acid-glycolic acid) (PLGA) NPs and research Naidoo et al. (2019) showed excellent active targeting subcellular uptake, with enhanced PDT treatment outcomes when conjugating ZnPcS4 PSs and an antibody PLGA gold NP de-livery carrier, within in vitro melanoma cancer cultured cells, suggesting that both carriers could potentially serve as bioactive models.
  • Studies by Simelane et al. (2021) and Montaseri et al. (2022), showed improved subcellular localization and enhanced ZnPcS4 PS PDT treatment outcomes within in vitro cultured colorectal cancer cells, when conjugating it to an antibody actively targeted PLGA loaded gold NP or loading it in core/shell Ag@mSiO2 NP with folic acid, respectively.
  • A study by Chota et al (2022), investigated the in vitro cell death mechanisms induced by Dicoma anomala root extract in combination with ZnPcS4 PS mediated PDT in A549 lung cancer cells. This combination therapy confirmed the cytotoxic and antiproliferative effects of Dicoma anomala extracts in monotherapy and in combination with ZnPcS4-mediated PDT, through apoptosis and the upregulation of p38, p53, Bax, caspase 3, 8, and 9 apoptotic proteins, suggesting that combinative therapies are worth exploring.

The below recent published articles have been included throughout this manuscript which make reference to the excellent research in vivo outcomes utilizing ZnPcS4 PS as a PDT PS for cancer:

  • Studies by Brozek-Pluska et al. (2020) analyzed the fluorescence / Raman signals of ZnPcS4 PS, within in vivo human normal versus cancerous colon tissue samples and noted that this PS had a lower affinity for normal tissues. Moreover, this same study reported that ZnPcS4 PS had a monomer region signal, confirming its ideal PS properties within PDT applications. Lastly, this study noted that lower concentrations of ZnPcS4 PS were capable of endoplasmic reticulum (ER) localization, with mid concentrations capable of mitochondria and / or Golgi apparatus lysosomes localizations, while at higher concentrations reported preferential localization in the nucleus of colon cancer tissues. This was of importance, since the nucleus is the largest cellular organelle, which stores genetic information and should PDT induced singlet oxygen be able to destroy it in cancer tissues, enhanced therapeutic outcomes can be achieved.
  • In a study performed by Portilho et al. (2013) an albumin nanosphere (AN) containing ZnPcS4 PSs was developed. Results reported its excellent PDT antitumor activity within in vivo Swiss albino mice using an Ehrlich solid tumor as an experimental model for breast cancer. Intratumorally the ZnPcS4-AN was capable of mediating PDT to refrain tumor aggressiveness, as well as induce regression. Moreover, the use of this ZnPcS4-AN mediating PDT exposed anti-neoplastic activity, like that obtained while using intratumoral conventional chemo-Dox therapy.
  • More recently, a study by Dias et al. (2022), investigated interstitially targeted liposomes (ITLs) encapsulating ZnPcS4 PSs in vivo to try bring this platform closer to clinical transition. The key findings from this study was that the ZnPcS4 PS did not elicit notable systemic toxicity in zebrafish and chicken embryos and in human tumor breast cancer xenografts it noted significant tumor killing capacitation, however it did report skin phototoxicity in mouse models. This study concluded that more effective and safer carrier delivery models must be developed to integrate this PS into a comprehensive tumor targeting and delivery platform, so that future in vivo research can possibly translate into clinical applications.

  • The authors of the article either do not know the subject of research, or deliberately distort it in order to enhance the significance of experiments with ZnPcS4 (see lines 62-67).

The authors take offense from this statement; it implies that you are suggesting that we are either unethical or that we are inferior researchers. Your explanation below was sufficient without being prejudicial and derogatory.  

  • I'll explain. Tetra-sulfonated phthalocyanines (no matter metalless, with zinc or aluminum) penetrate very weakly into cells due to a large negative charge and therefore have a low ability to photodynamically damage cancer cells. It is necessary to reduce the number of sulfogroups in phthalocyanine in order to enhance its penetration into cancer cells and its photocytotoxicity. Phthalocyanine with a number of sulfogroups less than 4 can be obtained only as a mixture of variously sulfonated derivatives: mono, di, tri, tetra. Therefore, researchers are trying to find the optimal balance and obtain a mixture of phthalocyanines enriched with the most active disulfonated derivatives, while maintaining the solubility of the mixture of phthalocyanines.
  • Taking into account this reality, authors should not cite the data about cytotoxicity or antitumor activity of low sulfonated phthalocyanines as a justification for selection of ZnPcS4. They should not speculate that improved solubility of ZnPcS4 is advantage for anticancer PDT.

Authors have extensively within the introduction of the paper elaborated upon the choice to utilize and investigate the tetra sulphonation of ZnPcS4 to improve its solubility and hydrophilic nature, as well as reduce it tendency to aggregate. Moreover, authors have substantiated its uptake in cells with full explanations and a large amount of supportive literature.

  • I see that the authors violate the traditional ethics of scientific publications by manipulating statements and references allegedly confirming them. Below are two examples. The same applies; your explanation below is once again sufficient without being prejudicial. The same applies, your explanation below is once again sufficient without being prejudicial; this was not done with intent.

Lines 50-58. Incorrect and unsupported statement: authors of [9] did not study either “triplet ROS yields” or “ maximum  penetration of light into the tissues” for Pc PSs.  Reference [9], has been replace with Sekkat et al 2011 doi: 10.3390/molecules17010098.

Lines 64-67. Incorrect and unsupported statement: authors of [11] did not perform any studies proving that  “four sulfhydryl/thiol acetate groups … enhance PS solubility and so promote enhanced tumor subcellular localization and overall PDT treatment outcomes”. This statement has been removed and replaced with other supportive literature to elaborate upon sulphonation to promote solubilization and tumor passive uptake.

  • It seems that all similar statements and their support by references should be carefully checked, but it is not a duty of reviewer, it is the duty of authors.

Similar statements have been checked by authors to ensure that there is no perception of “violation of traditional ethics”

  • Among other drawbacks of the manuscript are artificial results presented in so called Table 1 and partially in Table 2 (in fact, they are Figures, not Tables). Why do the membrane probe ICAM-1 and nuclear probe DAPI have similar distribution in cells (see especially Table 1)? Why do ZnPcS4 and nuclear probe DAPI have similar distribution in many cells, if it is known that ZnPcS4 accumulates in cytoplasm not in the nucleus?

ICAM is a membrane stain; DAPI is a nuclear stain; as cells are round so obviously there will be an overlap of images whereby the nucleus is stained and the actual membrane of the cell will be observed, we did note that these result were qualitative, and no subcellular organelles were stained so to distinguish exact intra cellular localization is impossible. Furthermore, a study by Brozek-Pluska et al. (2020), has been referenced and discussed in the introduction which noted that lower concentrations of ZnPcS4 PS were capable of endoplasmic reticulum (ER) localization, with mid concentrations capable of mitochondria and/or Golgi apparatus lysosomes localizations, while at higher concentrations re-ported preferential localization in the nucleus of colon cancer tissues.

  • Another major drawback is unreasonably long Discussion section containing extensive repetitions of data and statements from Introduction and Results sections.

The discussion section has been reduced to remove repletion of data and statements.                 

  • Apart of some typos, terminology used by authors, many statements and even figure legends require careful verification and edition. As examples see legends to figures 3-6, (line 366) “during subcellular tumor absorption” and (line 874) “Cy5 red auto fluorescent signals “.

This has been checked and corrected where applicable.

  • Section Materials and methods contains trivial descriptions, like (lines 798-799) to “make a stock concentration of 0.0005 M, 0.0006 g of ZnPcS4 powder was added to 1.25 ml of 0.001 M PBS” and does not contain essential details for flow cytometry measurements, parameters and statistics.

This has been checked and corrected where applicable. Statistical analysis is elaborated upon in Section 4.8. Statistical Analysis of the manuscript.

Reviewer 3 Report

The manuscript submitted by Abrahamse and co-workers described photodynamic therapy using water-soluble zinc phthalocyanine in combination with cannabidiol as anti-cancer therapy. This is an interesting study, however, some flaws are found and should be removed before final publication:

1. The chemical structure of studied materials should be added to main text.

2. Why did the authors not perform standard MTT(MTS) assays to estimate cytotoxicity?

3. The authors stated that combination therapy has a synergistic effect. Did they measure it according to the ChouThalay method?\

4. What is the influence of cannabidiol on the production of singlet oxygen by zinc phthalocyanine? Did the authors calculate quantum yields of the generation of singlet oxygen?

5. The authors stated that one of the mechanisms of uptake of studied phthalocyanine is the EPR effect. Is that effect only limited to nanomaterials? Or the authors claim that it is in nanometric form (lack of DLS and TEM analysis).

Author Response

REVIEWER 3

Thank you for your feedback in enhancing the quality of our manuscript submission. These have been addressed by the authors, per reviewer’s comment, and responded to below in red. All requested corrections have been indicated in red font in the revised manuscript. In addition to noting changes in reference numbers, due to inserts and figure number changes and a full English editorial has been done.

Comments and Suggestions for Authors

This is an interesting study; however, some flaws are found and should be removed before final publication:

  1. The chemical structure of studied materials should be added to main text.

The chemical structures of ZNPcS4 and CBD have been added in the methodology, as Figure 9.

  1. Why did the authors not perform standard MTT(MTS) assays to estimate cytotoxicity?

The LDH leakage assay is based on the release of the enzyme into the culture medium after cell membrane damage whereas the MTT assay is mainly based on the enzymatic conversion of MTT in the mitochondria. LDH cytotoxicity assay is capable of detecting low level damage to cell membrane which cannot detected using other methods. And since it does not damage healthy cells, the LDH cytotoxicity assay can be performed directly in the cell culture wells containing a mixed population of viable and damaged cells. Hence this was the preferred method of choice.

  1. The authors stated that combination therapy has a synergistic effect. Did they measure it according to the Chou−Thalay method?

Chou–Talalay's method is useful for analysing synergistic/additive or antagonistic behaviour between biologically active substances. The Chou–Talalay combination index method for the analysis of combined bioactives, according to our understanding, is only valid under the specific circumstances when all components in the mix have identical concentration exponents. However, by increasing our understanding of the combined effects, novel combinations of bioactives can be more rigorously analysed in future studies. Furthermore, for enhanced clarity (as recommended by other reviewers), the word synergistic has been removed and replaced with the term/s combined/in combination.

  1. What is the influence of cannabidiol on the production of singlet oxygen by zinc phthalocyanine? Did the authors calculate quantum yields of the generation of singlet oxygen?

CBD showed no influence in cell death or cytoxicity assays within ZNPcS4 PS outcomes; therefore, generation of  singlet oxygen was not measured. However, since this was novel study, future studies should consider this.

  1. The authors stated that one of the mechanisms of uptake of studied phthalocyanine is the EPR effect. Is that effect only limited to nanomaterials? Or the authors claim that it is in nanometric form (lack of DLS and TEM analysis).

No nanoparticles were used in this study and so DLS and TEM analysis could not be done. For clarity purposes authors have elaborated upon this statement within the introduction;  Studies by Pashkovskaya et al. (2007) and Montaseri et al. (2022), have noted that Pc complexes, such as anionic ZnPcS4 PSs, can efficiently bind to cancer tumor phospho-lipid membranes, through metal–phosphate coordination. Therefore, ZnPcS4 PSs are able to spontaneously accumulate in tumors through an enhanced permeability and retention (EPR) effect, due to the binding of its tetrasulfonated groups to cancer membranes (which have high phospholipid contents) and so allow for passively selective accumulation.

Reviewer 4 Report

Razlog et al reported the combinative PDT and CBD on cervical cancer cells. The study has been extensively done and reported some interesting results, however, there are some flows in the presentation of data and figures which must be significantly addressed before publication. Here are my specific comments:

1.       Add photosensitizer name in the title.

2.       LDH membrane integrity assay was done for photosensitizer and CBD separately, however, the same experiment using both in combination must be done.

3.       Why the 0.125 uM and 0.5 uM concentrations of ZnPcS4 and CBD, respectively were chosen? Did the authors check other combinations?

4.       All figure quality is not good for a publication, it's mostly looking like lab reports. Merge the figures 1, 2, and combination study figures into a single figure (make a, b and c). the gaps between two columns are too much, there is no y-axis why? Why horizontal grids are there? The axis labels are invisible. Overall, I would recommend making the figure nice, visible, and clear. All figures must be formatted accordingly. Merge figures as much as possible.

5.       Table 1 and Table 2 are not actually presented as tables. They are microscopy figures.

6.       In cell internalization study, the time of internalization must be reported. There is no quantitative analysis which is essential for the internalization assay. Do the experiment again and report one quantitative internalization graph at each hour for up to 6 h at least.

7.        In the FACS result section, FACS analysis graphs must be presented along with the grouped column graphs. This will add the reliability of the results to the readers.

8.       How authors confirmed that this is a synergistic effect, not an additive effect?

9.       I would recommend adding some more references:

·         ACS Appl. Mater. Interfaces 2020, 12, 16, 18309–18318

·         ACS Appl. Bio Mater. 2019, 2, 1, 349–361

·         Nat Commun 9, 5044 (2018).

·         Nanomedicine: Nanotechnology, Biology and Medicine 33, 102368

·         Journal of Photochemistry and Photobiology B: Biology,140, 2014, 365-373

10.   What ANOVA analysis and post hoc test were done to determine the significant differences?

Author Response

REVIEWER 4

Thank you for your feedback in enhancing the quality of our manuscript submission. These have been addressed by the authors, per reviewer’s comment, and responded to below in red. All requested corrections have been indicated in red font in the revised manuscript. In addition to noting changes in reference numbers, due to inserts and figure number changes.

Comments and Suggestions for Authors

The study has been extensively done and reported some interesting results, however, there are some flaws in the presentation of data and figures which must be significantly addressed before publication.

Here are my specific comments:

  1. Add photosensitizer name in the title.

The PS name has been added to the title, as recommended; “Cytotoxic Effects of Combinative ZnPcS4 Photosensitizer Photodynamic Therapy (PDT) and Cannabidiol (CBD) on a Cervical Cancer Cell Line”

  1. LDH membrane integrity assay was done for photosensitizer and CBD separately, however, the same experiment using both in combination must be done.

The LDH test was used to measure inhibitory concentration (ICD50) doses of CBD and ZnPcS4 PS separately; the flow cytometry cell death pathway analysis was used for both in combination, which in relation to cell death, the cytotoxicity of this novel study can be assumed.

  1. Why the 0.125 uM and 0.5 uM concentrations of ZnPcS4 and CBD, respectively were chosen? Did the authors check other combinations?

The authors analyzed five different concentrations of both ZnPcS4 and CBD. The final ICD50 concentrations that were chosen were have been explained in Section 3.1. HeLa ZnPcS4 PS ICD50 PDT & LDH and Section 3.2. HeLa CBD ICD50 Irradiation & LDH. Explanation for calculation is contained in Section  4.4. Individual ZnPcS4 PS PDT or CBD Irradiation LDH Cytoxicity Dose Response Assays.

  1. All figure quality is not good for a publication, it's mostly looking like lab reports. Merge the figures 1, 2, and combination study figures into a single figure (make a, b and c). the gaps between two columns are too much, there is no y-axis why? Why horizontal grids are there? The axis labels are invisible. Overall, I would recommend making the figure nice, visible, and clear. All figures must be formatted accordingly. Merge figures as much as possible.

Whilst the authors attempted to merge graphs, these become too complex and illegible. The quality of all figures have been improved upon for enhanced clarity, taking the above suggestions into account and 600DPI images have been supplied.

  1. Table 1 and Table 2 are not actually presented as tables. They are microscopy figures.

Corrected; both Table 1 and 2 have been corrected to Figures 3 and 4, and labelling been changed to figure legends.

  1. In cell internalization study, the time of internalization must be reported. There is no quantitative analysis which is essential for the internalization assay. Do the experiment again and report one quantitative internalization graph at each hour for up to 6 h at least.

These experiments were not possible on our fluorescent microscope with internalization assay measurements, we require a confocal microscope which we don’t have, and so only qualitative images were shown.

  1. In the FACS result section, FACS analysis graphs must be presented along with the grouped column graphs. This will add the reliability of the results to the readers.

In total there are 22 FACS analysis graphs which accompany the data represented as FACS bar graphs. Firstly due to space limitations on the amount of figures one can put in a manuscript to put all 22 in is impossible. However, authors are willing to be transparent with their data and all raw data will be provided should the reader wish to access it from the for the noted raw data repository link in the data availability statement once published.       

  1. How authors confirmed that this is a synergistic effect, not an additive effect?

For enhanced clarity (as recommended by other reviewers), the word synergistic has been removed from manuscript and replaced with the term/s combined/in combination.

  1. I would recommend adding some more references:
  • ACS Appl. Mater. Interfaces 2020, 12, 16, 18309–18318
  • ACS Appl. Bio Mater. 2019, 2, 1, 349–361
  • Nat Commun 9, 5044 (2018).
  • Nanomedicine: Nanotechnology, Biology and Medicine 33, 102368
  • Journal of Photochemistry and Photobiology B: Biology,140, 2014, 365-373

Thank you for referral to these references to improve the manuscript. The study by Zhu et al., (2020) relating to “A Dual Functional Nanoreactor for Synergistic Starvation and Photodynamic Therapy”; both studies by Thakur et al., (2019 & 2021) on “Self-Assembled Gold Nanoparticle-Lipid Nanocomposites for On-Demand Delivery, Tumor Accumulation, and Combined Photothermal-Photodynamic Therapy” and “Co-administration of zinc phthalocyanine and quercetin via hybrid nanoparticles for augmented photodynamic therapy”; and three references by Zhou et al (2022) have been added to the manuscript.

  1. What ANOVA analysis and post hoc test were done to determine the significant differences?

Statistical analysis has been explained and elaborate upon in Section 4.8: The student t-test and one-way analysis of variance (ANOVA) was used for comparing two or more means of normally distributed data; whereas the Mann-Whitney test was used for non-normally distributed data to determine the statistical difference between the various control and experimental groups. The error bars on graphs display the mean standard error (SEM) and if an experimental or control group reported a SD higher than 1, within a sample repeat, it was repeated to ensure data accuracy and precise statistical comparisons. Statistical differences between the various control and experimental groups were determined; values in the 95% confidence interval (P < 0.05*, P < 0.01** or P < 0.001***) were accepted as statistically different and represented graphically.

Reviewer 5 Report

The manuscript "Cytotoxic Effects of Combinative Photodynamic Therapy (PDT) 2 and Cannabidiol (CBD) on Cervical Cancer Cells presents an original research study which is continuation of the research studies from these authors.

Comments:

1. The text under the X-axis is hardly visible. Please improve the presentation.

2. line 50-51: "passive affinity uptake" Please clarify the sentence, because not every PS is selective.

3. line 64: "sulphide" means only "S" Maybe you mean sulphate?

4. Table 1 and Table 2 are not typical Tables, better to stay Figure ...

5. The magnification is not readable and should be added bellow in the text.

6. All the texts bellow the Figures should be improved (without last sentanse).

Author Response

REVIEWER 5

Thank you for your feedback in enhancing the quality of our manuscript submission. These have been addressed by the authors, per reviewer’s comment, and responded to below in red. All requested corrections have been indicated in red font in the revised manuscript. In addition to noting changes in reference numbers, due to inserts and figure number changes.

Comments and Suggestions for Authors

The manuscript "Cytotoxic Effects of Combinative Photodynamic Therapy (PDT) 2 and Cannabidiol (CBD) on Cervical Cancer Cells presents an original research study which is continuation of the research studies from these authors.

Comments:

  1. The text under the X-axis is hardly visible. Please improve the presentation.

Corrected; the quality of all the figures have been enhanced, and 600DPI images have been supplied.

  1. line 50-51: "passive affinity uptake" Please clarify the sentence, because not every PS is selective.

The sentence has been amended.

  1. 3. line 64: "sulphide" means only "S" Maybe you mean sulphate?

Corrected; to sulphate.

  1. Table 1 and Table 2 are not typical Tables, better to stay Figure ...

Corrected; both Table 1 and 2 have been corrected to Figures 3 and 4, and labelling been changed to figure legends.

  1. The magnification is not readable and should be added below in the text.

The magnification has been added in Figure 3 and 4 legends text and the scale bars of all images have been enhanced for clarity.

  1. All the texts bellow the Figures should be improved (without last sentence).

Corrected; the labelling of all the figures has been improved incorporating the last sentence.

Round 2

Reviewer 2 Report

Revised version of the manuscript was not improved sufficiently to recommend it publication in IJMS.

I am sure that the studied photosensitizer (ZnPcS4) is not suitable for the real photodynamic anticancer therapy, because it was already tested in the extended preclinical studies in vitro and in vivo by several research groups involved in the commercial development of drugs. Results of these studies were not published because of their negative character. They also revealed that coordination with Zn gives no advantage in vivo compared to coordination with Al as well as compared to metal-free sulfonated phthalocyanine.    

Authors of the manuscript demonstrates that insufficient photoinduced cytotoxicity of ZnPcS4 can be improved in vitro if it is combined with CBD treatment, but it is unclear, if this effect can be reproduced in vivo. Absence of supportive in vivo data reduce the significance of the results obtained in vitro.  Speculations about possible further improvement of photoinuced cytotoxicity of ZnPcS4 due to nanoencapsulation and/or targeting do not increase significance of the presented results.   

Authors continue violate the traditional ethics of scientific publications by manipulating statements and references allegedly confirming them. They are doing that even in the revised text, that should correct the previous violation. See, for example lines 57-59. Authors write “the incorporation of zinc (Zn2+) diamagnetic metal ions ensures that Pc PSs have even higher triplet ROS yields with a longer triplet lifetime as well as a maximum penetration of light into the tissues [9].  But in the reference [9], it is stated that “the presence of a diamagnetic central metal such as Zn2+ and Al3+ in the Pc nucleus seems to improve the triplet state life time, as well as its yield and singlet oxygen yields compared to paramagnetic metals… However, the metallation is not required for its photodynamic activity.” Feel the difference.  Here, even last statement about “maximum penetration of light into the tissues” is not correct, because light of 750-850 nm together with naphtalocyanines and bacteriochlorins provides considerably deeper photodynamic treatment of tumors.   

Authors should never cite  “preferential localization in the nucleus of colon cancer tissues [15]” , because nuclear localization of ZnPcS4 is artificial result. It arises when cells are damaged in the presence of tetrasulfonated phthalocyanines: tetrasulfonated phthalocyanines come from cytoplasm to the nucleus. In intact cells, ZnPcS4 does not penetrate inside nuclei. Authors can verify this themselves by studying living cells by either epifluorescence or confocal microscopy with high magnification and resolution.  

Major drawback is unreasonably long text in every part of the manuscript: Introduction, Results, Discussion, Materials and Methods sections. Text contains extensive repetitions of data, descriptions and statements.  Description of rather simple experiments and results is very bulky everywhere.     

Terminology used by authors and many statements require careful verification and edition:

“a monomer region signal”,  “clinical transition”, “tumor killing capacitation”, “cancer membranes”, “PDT laser irradiation”, “Lactate Dehydrogenase membrane damage integrity analysis”, “results were found in cells”, “the benchmark comparator”, “pre-PDT/irradiation”, “Cy5 fluorescence for ZnPcS4”, “Final Combinative Assays”, “high photochemical stability in the absence of light”, “Cy5 red fluorescent signals that were produced from the ZnPcS4”, “the late apoptosis induced cell death pathway”,  etc.

Presentation of experimental data and their errors is very strange: 3% (±SEM 0.47), 52%** (±SEM 0.75), etc.

There are many very strange sentences, for example:

“The determined ICD50 CBD concentration able to induce approximately 50% cytotoxicity within HeLa experimental groups was found to be 0.5 µM, since it reported 52%** cytotoxicity.”  

“It was able to accumulate within in vitro cultured CC HeLa cells more passively, when compared to normal WS1 fibroblast cells”.

Section Materials and methods still contains trivial descriptions, for example:

“To make a working stock concentration of 125 µM, 1 ml of 0.0005 M ZnPcS4 PS stock solution was 840 diluted with 4 ml of 0.001 M PBS.” “The CBD was 847 diluted with 19 ml of 99.8% of ethanol to make a stock concentration of 0.5 mg/ml.”

Data presented in Figures 3 and 4 were obtained for fixed cells and highly likely contain fixation-induced artifacts in distribution of ZnPcS4. These data were obtained and presented at very low resolution and cannot be used for any conclusions about cellular localization of ZnPcS4. Such data should be measured with confocal microscopy at high resolution using living (not fixed) cells.   

All the essential parameters of optical microscopy and flow cytometry measurements should be briefly indicated in the Materials and Methods section.     

Finally, I conclude that the presented manuscript is not ready for publication in highly rated journals like IJMS.

Author Response

Reviewer 2 Comments and Suggestions Authors Response

  • Revised version of the manuscript was not improved sufficiently to recommend it publication in IJMS.
    • We have responded, corrected and included ALL changes as per your recommendations and request.
  • I am sure that the studied photosensitizer (ZnPcS4) is not suitable for the real photodynamic anticancer therapy, because it was already tested in the extended preclinical studies in vitroand in vivo by several research groups involved in the commercial development of drugs. Results of these studies were not published because of their negative character. They also revealed that coordination with Zn gives no advantage in vivo compared to coordination with Al as well as compared to metal-free sulfonated phthalocyanine.    
    • A review of the literature that is in the public domain clearly demonstrates the value of this photosensitizer and any clinical trial that has not been published does not diminish the PSs potential. Also data from clinical trials cannot be compared directly to in vitro research. According to previous supplied literature as per your former request, ZnPcS4 is a suitable photosensitizer according to PUBLISHED literature.
  • Authors of the manuscript demonstrates that insufficient photoinduced cytotoxicity of ZnPcS4 can be improved in vitroif it is combined with CBD treatment, but it is unclear, if this effect can be reproduced in vivo. Absence of supportive in vivo data reduce the significance of the results obtained in vitro.  Speculations about possible further improvement of photoinduced cytotoxicity of ZnPcS4 due to nanoencapsulation and/or targeting do not increase significance of the presented results.
    • The current manuscript is showing the research we have conducted and notes that research will remain ongoing to possible in vivo studies etc. just as all investigations in research are ongoing for future development, and so authors suggest leaving the final decision to the editor.  
  • Authors continue violate the traditional ethics of scientific publications by manipulating statements and references allegedly confirming them. They are doing that even in the revised text, that should correct the previous violation.
    • This comment is highly offensive. We are trying to indicate and provide evidence required by you but cannot quote directly from previously published work. Hence we are writing in our own words.
  • See, for example lines 57-59. Authors write “the incorporation of zinc (Zn2+) diamagnetic metal ions ensures that Pc PSs have even higher triplet ROS yields with a longer triplet lifetime as well as a maximum penetration of light into the tissues [9].  But in the reference [9], it is stated that “the presence of a diamagnetic central metal such as Zn2+ and Al3+ in the Pc nucleus seems to improve the triplet state lifetime, as well as its yield and singlet oxygen yields compared to paramagnetic metals… However, the metallation is not required for its photodynamic activity.” Feel the difference.
    • Sentence has been rephrased to directly quote reference.
  • Here, even last statement about “maximum penetration of light into the tissues” is not correct, because light of 750-850 nm together with naphtalocyanines and bacteriochlorins provides considerably deeper photodynamic treatment of tumors.   
    • Statement removed.
  • Authors should never cite “preferential localization in the nucleus of colon cancer tissues [15]” because nuclear localization of ZnPcS4 is artificial result. It arises when cells are damaged in the presence of tetrasulfonated phthalocyanines: tetrasulfonated phthalocyanines come from cytoplasm to the nucleus. In intact cells, ZnPcS4 does not penetrate inside nuclei. Authors can verify this themselves by studying living cells by either epifluorescence or confocal microscopy with high magnification and resolution.  
    • This was quoted directly from given published reference that is in the public domain, and so again above statement is of your opinion and not constructive criticism.
  • Major drawback is unreasonably long text in every part of the manuscript: Introduction, Results, Discussion, Materials and Methods sections. Text contains extensive repetitions of data, descriptions, and statements.  Description of rather simple experiments and results is very bulky everywhere.
    • Again, of your opinion this manuscript has been reviewed by 4 other reviewers, excluding yourself and none of them have commented on any of these aspects, and accepted the manuscript for publication with minor corrections.
    • In addition, authors feel de-bulking the manuscript any further will compromise its clear flow and understanding and so recommend leaving this to the editor’s choice.
  • Terminology used by authors and many statements require careful verification and edition: “a monomer region signal”,  “clinical transition”, “tumor killing capacitation”, “cancer membranes”, “PDT laser irradiation”, “Lactate Dehydrogenase membrane damage integrity analysis”, “results were found in cells”, “the benchmark comparator”, “pre-PDT/irradiation”, “Cy5 fluorescence for ZnPcS4”, “Final Combinative Assays”, “high photochemical stability in the absence of light”, “Cy5 red fluorescent signals that were produced from the ZnPcS4”, “the late apoptosis induced cell death pathway”,  etc.
    • “a monomer region signal”: quoted directly from reference.
    • “clinical transition”: changed to “clinical investigations”.
    • “tumor killing capacitation” changed to “significant tumor reduction”.
    • “cancer membranes” changed to “cancer cell membranes”.
    • “PDT laser irradiation”, PDT has been removed and throughout manuscript referred to as laser irradiation.
    • “Lactate Dehydrogenase membrane damage integrity analysis”, full name of test analysis and kit that was used so can’t be changed.
    • “results were found in cells”: sentence corrected to “Similar overall average cytoxicity results were reported for control groups consisting of cells plus CBD of ±53%**”.
    • “the benchmark comparator” changed to control comparator.
    • “pre-PDT/irradiation” changed to pre-irradiation.
    • “Cy5 fluorescence for ZnPcS4” changed to Cy5 fluorescence from ZnPcS4.
    • “Final Combinative Assays”: corrected throughout the manuscript was instead of stating “Final Combinative Assays” authors have referred to them as combinative assays and excluded the word final.
    • “high photochemical stability in the absence of light” changed to “high stability in the absence of light”
    • “the late apoptosis induced cell death pathway”: sentence corrected to “determine that ZnPcS4 PS and CBD in cultured HeLa CC cells, was cable of inducing late apoptotic forms of cell death”.
    • “Cy5 red fluorescent signals that were produced from the ZnPcS4” changed to “Cy5 red fluorescent signals that were produced by the ZnPcS4
    • “the late apoptosis induced cell death pathway” removed and sentence clarified: as well as determine that ZnPcS4 PS and CBD in cultured HeLa CC cells, was cable of inducing late apoptotic forms of cell death,
  • Presentation of experimental data and their errors is very strange: 3% (±SEM 0.47), 52%** (±SEM 0.75), etc.
    • The experimental data standard error mean has been checked and this is correct, please remember SEM is influenced by standard deviation and number of repeats, so we are unsure of what you mean by “strange”.
  • There are many very strange sentences, for example: “The determined ICD50 CBD concentration able to induce approximately 50% cytotoxicity within HeLa experimental groups was found to be 0.5 µM, since it reported 52%** cytotoxicity.”  “It was able to accumulate within in vitro cultured CC HeLa cells more passively, when compared to normal WS1 fibroblast cells”.
    • Both above sentences have been corrected for clarity in the manuscript.
  • Section Materials and methods still contains trivial descriptions, for example: “To make a working stock concentration of 125 µM, 1 ml of 0.0005 M ZnPcS4 PS stock solution was 840 diluted with 4 ml of 0.001 M PBS.” “The CBD was 847 diluted with 19 ml of 99.8% of ethanol to make a stock concentration of 0.5 mg/ml.”
    • Please be advised that 4 additional reviewers requested minor changes and while they are asking for detail, you are criticizing it. Again, of your opinion this manuscript has been reviewed by 4 other reviewers, excluding yourself and none of them have commented on any of these aspects, and accepted the manuscript for publication.
  • Data presented in Figures 3 and 4 were obtained for fixed cells and highly likely contain fixation-induced artifacts in distribution of ZnPcS4. These data were obtained and presented at very low resolution and cannot be used for any conclusions about cellular localization of ZnPcS4. Such data should be measured with confocal microscopy at high resolution using living (not fixed) cells.   
    • Noted and the sections in the manuscript referring to subcellular localization have been clarified to state that the results are qualitative and so cannot be utilized to make concrete conclusions, but rather suggestive findings and future confocal microscopy should be considered.
  • All the essential parameters of optical microscopy and flow cytometry measurements should be briefly indicated in the Materials and Methods section.     
    • Optical microscopy parameters have already been mentioned in the original submitted manuscript within the materials and methods section: “The slides were examined using the filter settings of a Carl Zeiss Axio Z1 Observer immunofluorescent microscope at 40X magnification. The 358Ex/461Em filter was used to detect blue DAPI counter stained nuclei in cultured cells, while the 495Ex/519Em filter was used to detect any green FITC stained ICAM-1 membrane proteins in cultured cells. Lastly, the 589Ex/610Em filter was used to detect any Cy5 red fluorescent signals that were produced from the ZnPcS4 PS within the various control and experimental groups, to determine if the PS was capable of subcellular localization in HeLa cells only and had no effective uptake in WS1 cells, when combined with CBD”.
    • BD Accuri™ C6 flow cytometer essential parameters have been elaborated upon with the materials and methods section of the manuscript, as well as FACS gating strategies for live, early/ late apoptosis and necrosis have been supplied as supplementary information.
  • Finally, I conclude that the presented manuscript is not ready for publication in highly rated journals like IJMS.
    • Noted and we would like to kindly request if you could leave the final decision to the editor, since 4 other reviewers, excluding yourself are in favor of this manuscript’s publication.

Reviewer 3 Report

I suggest acceptance of the current version of the manuscript. The authors made a proper correction.

Author Response

Reviewer 3 Comments and Suggestions Authors Response:

  • I suggest acceptance of the current version of the manuscript. The authors made a proper correction.
    • Noted and thank you for your valuable input and review of this manuscript.

Reviewer 4 Report

The authors tried well to answer my question, however, I do not agree with the authors' responses to questions no 6 and 7. 

Authors can use ImageJ software for quantification. Or authors can use FACS for the study and report.

For question 7, the facs figures can be added in the supporting info. 

After these minor changes, I would recommend this manuscript for publication. 

Author Response

Reviewer 4 Comments and Suggestions Authors Response

  • The authors tried well to answer my question; however, I do not agree with the authors' responses to questions no 6 and 7. 
  • Authors can use ImageJ software for quantification.
    • Since, the images in Figures 3 and 4 were captured utilizing fluorescent microscopy with fixed cells, there is a possibility that they could contain fixation-induced artifacts in distribution of ZnPcS4 PS, which can affect their overall resolution and so they can only be utilized for quantitative observations with hypothetical conclusions. These images are not suitable to be utilized and quantified with ImageJ software. Thus, it has been corrected throughout the manuscript to refer to subcellular localization assays as merely observation with suggestive outcomes and authors have noted in the manuscript that further confirmatory confocal microscopy studies with live cells needs to be performed and measured to confirm the possible sub cellular localization abilities of ZnPcS4
  • Or authors can use FACS for the study and report. For question 7, the facs figures can be added in the supporting info. 
    • BD Accuri™ C6 flow cytometer essential parameters have been elaborated upon with the materials and methods section of the manuscript, as well as FACS gating strategies for live, early/ late apoptosis and necrosis being supplied.
  • After these minor changes, I would recommend this manuscript for publication. 
    • Noted and thank you for your valuable input and review of this manuscript.

Round 3

Reviewer 2 Report

-